# TRAM: Test-time Risk Adaptation with Mixture of Agents

## Abstract

Deployed reinforcement learning agents must satisfy safety requirements that emerge only at test time—evolving regulations, unexpected hazards, or shifted operational priorities. Current risk-aware methods embed fixed risk models (typically return variance) during training, but this approach suffers from two fundamental limitations: it restricts risk expressiveness to trajectory-level statistics, and it induces uniform conservatism that reduces behavioral coverage needed for effective deployment adaptation. We propose **TRAM** (**T**est-time **R**isk **A**daptation via **M**ixture of **A**gents), a deployment-time framework that composes risk-neutral source policies to satisfy arbitrary risk specifications without retraining. TRAM represents risk through occupancy-based functionals that capture spatial constraints, behavioral drift, and local volatility—risk types that trajectory variance cannot encode. Our theoretical analysis provides localized performance bounds that cleanly separate reward transfer quality from risk alignment costs, and proves that risk-neutral source training is minimax optimal for deployment risk adaptation. Empirically, TRAM delivers superior safety-performance trade-offs across gridworld, continuous control, and large language model domains while maintaining computational efficiency through successor feature implementation.

## 1 Introduction

Reinforcement learning has achieved remarkable performance in controlled settings, yet deployed agents remain vulnerable to unforeseen conditions. Policies trained in simulation or narrow regimes often fail when facing new deployment constraints—unexpected pedestrian behaviors, sensor malfunctions, or updated safety regulations in autonomous driving exemplify this brittleness. These failures represent more than performance degradation; they constitute *deployment-time risks* that current RL systems cannot adapt to at test time. The fundamental challenge is that deployment risks differ systematically from training surrogates: dynamics shift, constraints emerge, and safety priorities evolve. A warehouse robot optimized for throughput may later face speed limits or spatial restrictions imposed by updated safety policies. Such risks are dynamic, diverse, and often unknown during training, making methods that assume fixed reward and risk models fragile in practice.

**Why training-time risk modeling is insufficient.** Conventional risk-sensitive RL addresses uncertainty by embedding specific risk objectives (variance penalties, CVaR constraints) directly into training procedures Gimelfarb et al. (2021). This approach presupposes that deployment risks are both *known a priori* and *stationary over time*. When risk profiles shift or are misspecified—common occurrences in real-world deployment—the learned policy becomes misaligned with actual safety requirements. Moreover, retraining for every new risk variant is impractical or unsafe in critical domains like robotics, healthcare, and autonomous systems. Training-time risk modeling thus represents a fundamental mismatch with deployment realities where safety requirements evolve continuously.

**The need for test-time risk adaptation.** To operate safely under shifting or previously unknown hazards, agents must adapt their risk-performance trade-offs *at deployment time*. Rather than committing to a single risk notion during training, agents should evaluate and synthesize behaviors using *deployment-specific* safety constraints—without any additional training or parameter updates. This capability enables dynamic safety adaptations that respect real-time operational constraints while preserving task performance.

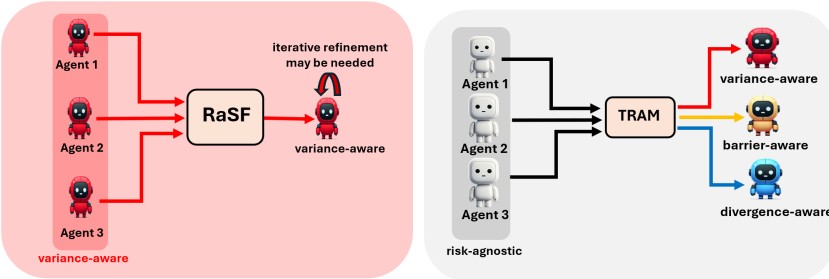

Figure 1: Comparison between existing risk-aware adaptation (RaSF Gimelfarb et al. (2021), top) and TRAM (bottom). TRAM eliminates the need for risk-aware source training, supports general occupancy-based risk models beyond variance, and synthesizes deployment policies through test-time composition without parameter updates.

**Our approach: TRAM.** We introduce **TRAM** (**T**est-time **R**isk **A**daptation via **M**ixture of **A**gents), a test-time framework that composes *risk-neutral* source policies to satisfy deployment-time risk objectives. TRAM requires no fine-tuning and makes no commitment to specific risk models during training. Instead, it evaluates each source policy on the target task and selects actions by maximizing risk-adjusted scores that combine expected return with occupancy-based risk assessments.

Critically, TRAM supports *general risk functionals* $\rho$, possibly defined over state-action occupancy measures $d$, including spatial hazards (barrier risk), behavioral constraints (KL divergence from safe references), and temporal volatility (per-step variance). This occupancy-based formulation captures domain-specific safety concerns that trajectory-level statistics cannot represent, as demonstrated in our theoretical analysis and empirical validation across discrete and continuous domains.

**Contributions.**

1. **Test-time risk adaptation framework.** We propose TRAM, which constructs risk-aware deployment policies by composing risk-neutral source agents using general risk functionals. The approach requires no retraining and enables adaptation to previously unseen risk specifications at deployment time.
2. **Expressive occupancy-based risk modeling.** We demonstrate that occupancy measures $d^\pi(s, a)$ enable risk specifications that capture spatial hazards, behavioral constraints, and temporal patterns that trajectory-level statistics cannot represent. Our framework supports diverse risk types including barrier avoidance, divergence control, and local volatility through unified occupancy-based functionals.
3. **Theoretical foundations and optimality results.** We derive localized performance bounds that decompose approximation error into reward transfer quality and risk alignment costs, revealing when and why TRAM succeeds. We prove that risk-neutral source training is minimax optimal for worst-case deployment risk, providing theoretical justification for TRAM's design.
4. **Computational efficiency and empirical validation.** TRAM achieves computational efficiency through successor feature implementation, enabling one-shot value evaluation via dot products. Empirical evaluation across gridworld, continuous control, and large language model domains demonstrates improved safety-performance trade-offs while maintaining deployment-time computational feasibility.

## 2 RELATED WORK

**Zero-shot and test-time transfer.** Zero-shot RL methods learn shared representations for immediate generalization without finetuning Marom & Rosman (2018); Oh et al. (2017); Higgins et al. (2017); Rezaei-Shoshtari et al. (2023); Touati et al. (2022). While effective for reward transfer, these approaches assume fixed evaluation objectives and provide no mechanism to *inject deployment-time risk* at inference. TRAM explicitly incorporates user-specified risk models at test time.

**Risk-sensitive RL.** Classical risk-aware RL optimizes training objectives augmented with variance, CVaR, or related criteria Bisi et al. (2019); Fei et al. (2020); Jain et al. (2021b); Mannor & Tsitsiklis (2013); Mao et al. (2018); Nass et al. (2019); Shen et al. (2014); Tamar et al. (2016); Whiteson (2021).

Table 1: Comparison of TRAM with representative families. Columns: risk sensitivity, support for risk types beyond variance, cross-task transfer, use of shared task structure, and little/no test-time compute.

| METHOD | RISK-AWARE | GENERAL RISK | TRANSFER | TASK STRUCT. | TEST-TIME |
|---|---|---|---|---|---|
| Standard RL | ✓ | ✗ | ✗ | ✗ | ✗ |
| Zero-shot RL | ✗ | ✗ | ✓ | ✗ | ✓ |
| Dual RL | ✓ | ✓ | ✗ | ✗ | ✗ |
| Risk-aware Adaptation | ✓ | ✗ | ✓ | ✗ | ✗ |
| **TRAM (ours)** | ✓ | ✓ | ✓ | ✓ | ✓ |

**Refs. Standard RL** Bisi et al. (2019); Fei et al. (2020); Jain et al. (2021b); Mannor & Tsitsiklis (2013); Mao et al. (2018); Nass et al. (2019); Shen et al. (2014); Tamar et al. (2016); Whiteson (2021); **Zero-shot RL** Marom & Rosman (2018); Oh et al. (2017); Higgins et al. (2017); Rezaei-Shoshtari et al. (2023); Touati et al. (2022); **Dual RL** Zhang et al. (2021); **Risk-aware Adaptation/SFs** Gimelfarb et al. (2021); Turchetta et al. (2020); Srinivasan et al. (2020); Held et al. (2017); García & Fernández (2019); Mankowitz et al. (2016); Jain et al. (2021a); Mankowitz et al. (2018); Barreto et al. (2017; 2018; 2020).

These approaches improve robustness when risk is known a priori, but hard-code risk at training and require retraining when deployment risks shift. Dual formulations broaden risk signals Zhang et al. (2021), yet still entail solving optimization problems tailored to specific risks at test time—often infeasible under tight latency.

**Risk-aware adaptation and safety transfer.** A complementary line learns teachers/critics for safe adaptation Turchetta et al. (2020); Srinivasan et al. (2020), robotics-oriented safety transfer Held et al. (2017), probabilistic policy reuse with risk García & Fernández (2019), or hierarchical controllers with risk-sensitive skills/options Mankowitz et al. (2016); Jain et al. (2021a); Mankowitz et al. (2018). These methods often assume a particular risk form and/or require nontrivial computation during adaptation.

**Successor features and RaSF.** Successor features (SFs) Barreto et al. (2017; 2018; 2020) exploit shared dynamics to evaluate policies across reward variants via dot products, enabling efficient transfer. Risk-aware SFs (RaSF) Gimelfarb et al. (2021) extend this idea to mean–variance, but (i) require *risk-aware* source training and (ii) remain bound to a narrow risk proxy (return variance). TRAM differs on both axes: (a) sources are explicitly *risk-neutral*, which we prove is minimax-optimal for worst-case Lipschitz deployment risks, and (b) deployment risk is an *arbitrary convex mixture* of occupancy-based functionals (e.g., barrier sets, KL-to-reference, per-step volatility), handled in a single pass without retraining.

**Positioning.** TRAM occupies the intersection of zero-shot transfer and risk-aware decision-making: it composes risk-neutral sources at inference, injects occupancy-based risks without retraining, and provides localized suboptimality guarantees plus a minimax argument for risk-neutral sources.

## 3 PROBLEM FORMULATION

### 3.1 DEPLOYMENT SETTING

**Deployment-time safety as constrained optimization.** We model agent-environment interaction as an MDP $M = (\mathcal{S}, \mathcal{A}, p, R, \gamma)$ with state space $\mathcal{S}$, action space $\mathcal{A}$, transition kernel $p(\cdot \mid s, a)$, rewards $R(s, a, s')$, and discount $\gamma \in [0, 1)$. A policy $\pi$ induces action-value function $Q^\pi(s, a) = \mathbb{E}^\pi[G_t \mid S_t = s, A_t = a]$ where $G_t = \sum_{i=0}^{\infty} \gamma^i R_{t+i+1}$ is the discounted return.

At deployment, the agent faces target task $M_\mathrm{T} = (\mathcal{S}, \mathcal{A}, p, R_\mathrm{T}, \gamma)$ with performance measured by expected return $J(\pi) = \mathbb{E}_\pi[\sum_{t \geq 0} \gamma^t R_\mathrm{T}(S_t, A_t, S_{t+1})]$. Crucially, deployment also introduces safety constraints specified only at test time through risk functional $\rho(\pi)$, yielding the constrained optimization problem:

$$\max_\pi J(\pi) \quad \text{subject to} \quad \rho(\pi) \leq \delta \tag{3.1}$$

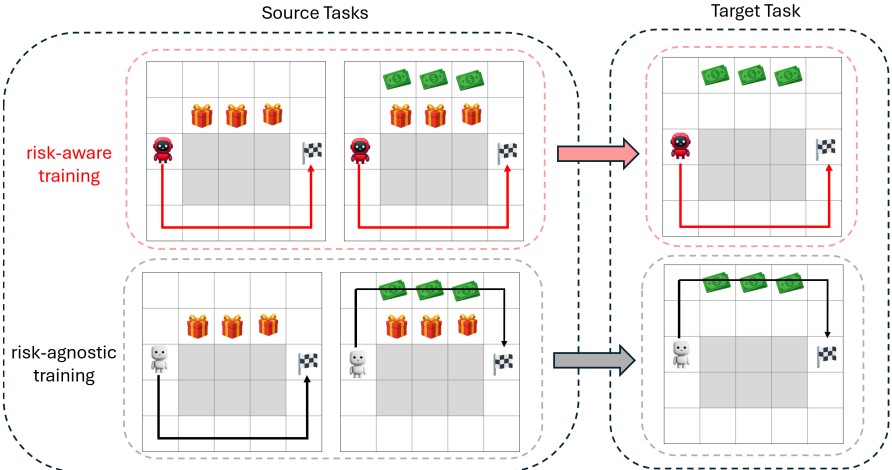

Figure 2: **Impact of risk-aware versus risk-neutral source training.** Two source policies (left) transfer to a target task (right). The **gift** symbol represents stochastic rewards (bomb or cash with equal probability). **Top:** Risk-aware training produces identical conservative sources that avoid the upper path, causing the deployed policy to miss the now-optimal route. **Bottom:** Risk-neutral training yields diverse sources enabling successful adaptation to the safer, higher-return upper path.

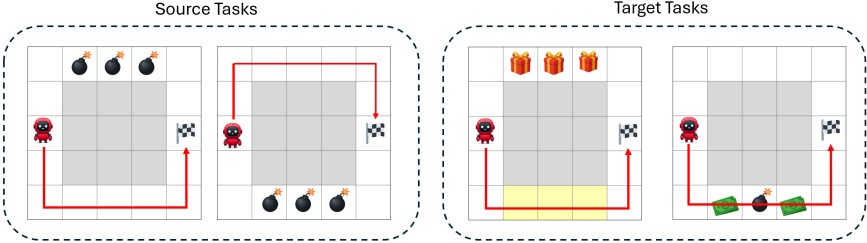

Figure 3: **Failure modes of trajectory return variance. Left:** Deterministic dynamics produce zero return variance despite volatile per-step rewards. **Middle:** High per-step reward variability creates genuine risk that trajectory variance cannot detect. **Right:** Low-variance path crosses danger zone (yellow) while safer high-variance alternative exists.

where $\delta > 0$ is the risk tolerance. Since direct constraint enforcement during test-time adaptation is challenging, we optimize the penalized surrogate:

$$\max_{\pi} \tilde{J}_c(\pi) = J(\pi) - c\,\rho(\pi), \quad c \geq 0 \qquad (3.2)$$

**Transfer setting: no retraining allowed.** Training from scratch on the target task is impractical at deployment due to sample complexity and time constraints. Instead, we assume access to pre-trained source policies $\{\pi_j\}_{j=1}^n$ from related source tasks $M_j = (\mathcal{S}, \mathcal{A}, p, R_j, \gamma)$ that share dynamics $p$ but differ in rewards. The challenge is composing these fixed policies to approximately solve Eq. 3.2 without parameter updates.

### 3.2    WHY VARIANCE-BASED RISK FAILS AT DEPLOYMENT

Current risk-aware adaptation methods rely primarily on trajectory return variance Gimelfarb et al. (2021), optimizing mean-variance objectives. This approach suffers from three fundamental limitations that become apparent in deployment scenarios.

**Limitation 1: Conservative training destroys behavioral diversity.** When source policies are trained with variance penalties, they converge to uniformly conservative behaviors that avoid regions with reward stochasticity (Figure 2, top). This reduces the behavioral repertoire available for test-time composition, preventing recovery of high-performing policies even when they satisfy deployment-time constraints. Risk-neutral training preserves diverse behaviors across the performance-risk spectrum (bottom), enabling adaptive policy selection based on actual deployment conditions.

**Limitation 2: Variance degenerates in deterministic settings.** With deterministic dynamics and policies, trajectory return variance equals zero regardless of per-step reward fluctuations (Figure 3, left-middle). Variance-based selection degenerates to risk-agnostic behavior, providing no safety guidance even when clear hazards exist. This failure occurs precisely when deployment systems most need risk awareness—in predictable environments where step-level risks accumulate.

**Limitation 3: Variance misrepresents domain-specific risks.** Trajectory variance captures outcome uncertainty but ignores spatial hazards, behavioral constraints, or worst-case scenarios critical for deployment safety (Figure 3, right). A path with low return variance can traverse dangerous regions while a safer alternative exhibits higher variance due to benign stochasticity. Real deployment risks—collision avoidance, toxic content generation, resource constraints—require risk models that directly encode domain-specific safety concerns.

**The need for expressive deployment-time risk modeling.** These limitations reveal a fundamental mismatch between trajectory variance and deployment-time safety requirements. Effective risk-aware adaptation requires: (1) **diverse source behaviors** preserved through risk-neutral training, and (2) **expressive risk models** that capture domain-specific safety constraints at deployment time.

## 4  METHOD: TRAM—TEST-TIME RISK ALIGNMENT VIA MULTI-AGENT COMPOSITION

The limitations in Section 3.2 motivate TRAM's approach: preserve behavioral diversity through risk-neutral source training while enabling expressive risk modeling at deployment time.

**Occupancy-based risk modeling.**  TRAM represents risk as functionals over state-action occupancy measures $d^\pi(s,a)$—the normalized discounted visitation frequencies under policy $\pi$. This captures three essential risk categories: **Spatial hazards** via barrier risk $\rho_{\text{barrier}}(d) = -\log(\tau - d(\overline{\mathcal{S}}))$, penalizing occupancy in danger zones $\overline{\mathcal{S}}$ with smooth penalties that intensify as visitation approaches tolerance $\tau$. **Local volatility** via per-step variance $\rho_{\text{variance}}(d) = \mathbb{E}^d[(r - \mathbb{E}^d[r])^2]$, measuring step-level reward fluctuations that trajectory variance misses. **Behavioral drift** via divergence risk $\rho_{\text{KL}}(d) = \text{KL}(d\|\bar{d})$, maintaining proximity to safe reference behaviors $\bar{d}$ to prevent reward hacking or unsafe policy drift. These functionals integrate directly into the penalized deployment objective (Eq. 3.2) while remaining independent of source training.

**TRAM policy construction.** Given risk-neutral source policies $\{\pi_j\}_{j=1}^n$ and target task $M_T$, TRAM evaluates each source on the target to obtain both performance $Q_T^{\pi_j}(s,a)$ and risk $\rho(d^{\pi_j})$. The deployment policy combines these through risk-adjusted values:

$$\tilde{Q}_T^{\pi_j}(s,a) = Q_T^{\pi_j}(s,a) - c\,\rho(d^{\pi_j}), \quad c \geq 0 \tag{4.1}$$

Actions are selected by maximizing across all sources:

$$\pi_T(a \mid s) \in \arg\max_{b \in \mathcal{A}} \max_{j \in [n]} \tilde{Q}_T^{\pi_j}(s,b) \tag{4.2}$$

This directly operationalizes our penalized surrogate without any parameter updates, enabling immediate deployment adaptation.

### 4.1  WHEN AND WHY TRAM WORKS: THEORETICAL FOUNDATIONS

TRAM's theoretical analysis reveals fundamental insights about test-time risk adaptation: how approximation quality decomposes into interpretable components and why risk-neutral source training is provably optimal for deployment flexibility.

**The power of localized analysis.** Traditional transfer learning bounds require global reward similarity across all state-action pairs, often yielding pessimistic guarantees. TRAM's design enables much finer analysis through trajectory-specific mismatch terms that capture what actually matters—how well rewards align along paths that policies follow.

---

**Algorithm 1** TRAM: Test-time Risk Adaptation via Mixture of Agents

---

**Require:** Source policies $\{\pi_j\}_{j=1}^n$; risk weight $c \geq 0$; risk functional $\rho$
1: **for all** $(s, a) \in \mathcal{S} \times \mathcal{A}$ **do**
2:     **for** $j = 1$ to $n$ **do**
3:         Evaluate $Q_T^{\pi_j}(s, a)$ on target task $M_T$
4:         Compute risk $\rho(d^{\pi_j})$
5:     **end for**
6:     $a^* \in \arg\max_{b \in \mathcal{A}} \max_{j \in [n]} \left( Q_T^{\pi_j}(s, b) - c\,\rho(d^{\pi_j}) \right)$
7:     Set $\pi_T(a^* \mid s) = 1$, all others to 0
8: **end for**
**Ensure:** Deployment policy $\pi_T$

---

For any policy $\pi$ starting from $(s, a)$, define the localized reward discrepancy:

$$\Delta_\pi^{(T,j)}(s, a) = \sum_{t=0}^\infty \gamma^t \, \mathbb{E}_\pi \left[ |r_T(S_t, A_t) - r_j(S_t, A_t)| \mid S_0 = s, A_0 = a \right] \tag{4.3}$$

This measures cumulative reward mismatch along trajectories the policy actually visits, not across the entire state space.

> **Theorem 4.1** (Localized performance guarantee). *Let $\rho$ be L-Lipschitz in occupancy. For TRAM policy $\pi_T$ and risk-neutral optimal policy $\pi^*$ on the target task:*
>
> $$\tilde{Q}^{\pi^*}(s, a) - \tilde{Q}^{\pi_T}(s, a) \leq \min_{j \in [n]} \left\{ \Delta_{\pi^*}^{(T,j)}(s, a) + 2Lc \right\} \tag{4.4}$$
>
> *where $\tilde{Q}^\pi(s, a) = Q^\pi(s, a) - c\,\rho(d^\pi)$ are risk-adjusted values.*

**Key insight: Clean error decomposition.** This bound reveals TRAM's fundamental advantage—the approximation error separates cleanly into two interpretable components. The transfer quality $\Delta_{\pi^*}^{(T,j)}(s, a)$ depends only on how well the best source task matches the target along trajectories the optimal policy actually follows. The risk alignment cost $2Lc$ quantifies the price of imposing safety constraints at deployment time. This decomposition enables independent analysis: improve transfer quality by diversifying sources, control safety costs by tuning $c$.

**From risk-adjusted to standard performance.** Many practitioners care about unpenalized performance even when deploying with safety constraints. TRAM's risk-adjusted guarantees translate directly to standard Q-value bounds:

> **Corollary 4.2** (Standard performance bound). *Under the assumptions of Theorem 4.1:*
>
> $$Q^{\pi^*}(s, a) - Q^{\pi_T}(s, a) \leq \min_{j \in [n]} \left\{ \Delta_{\pi^*}^{(T,j)}(s, a) \right\} + 4Lc \tag{4.5}$$

**Key insight: The cost of safety.** The factor $4Lc$ precisely quantifies the total performance sacrifice for safety.

**Why risk-neutral sources are provably optimal.** A natural question is whether source policies should incorporate some risk awareness during training. We prove this intuition is wrong:

> **Theorem 4.3** (Minimax optimality of risk-neutral training). *Let $S_\lambda$ denote sources trained with risk weight $\lambda \geq 0$. For any class of L-Lipschitz deployment risks and regret metric $\mathcal{E}_{\rho,V}(\cdot)$:*
>
> $$\sup_\rho \mathcal{E}_{\rho,V}(S_0) = \inf_{\lambda \geq 0} \sup_\rho \mathcal{E}_{\rho,V}(S_\lambda) \tag{4.6}$$

**Key insight: Maximum deployment flexibility.** Here the regret $\mathcal{E}_{\rho,V}(S_\lambda)$ measures the gap between the best achievable deployment risk under a performance floor $V$ and the best risk attainable using

the convex hull of source occupancies $\text{co}(S_\lambda)$, i.e.,

$$\mathcal{E}_{\rho,V}(S_\lambda) = \inf_{\substack{d \in \mathcal{D} \\ J_{r_\text{T}}(d) \geq V}} \rho(d) - \inf_{\substack{d \in \text{co}(S_\lambda) \\ J_{r_\text{T}}(d) \geq V}} \rho(d),$$

with $\mathcal{D}$ the set of feasible occupancies and $J_{r_\text{T}}$ the target return. Risk-neutral training ($\lambda = 0$) minimizes this worst-case regret across all Lipschitz deployment risks, while adding risk penalties during source training ($\lambda > 0$) contracts the reachable occupancy space and strictly worsens adversarial performance. This provides theoretical justification for TRAM's design choice to defer all risk considerations to test time, preserving maximum adaptability to unknown or evolving safety requirements.

## 4.2 PRACTICAL IMPLEMENTATION: SUCCESSOR FEATURES

TRAM's deployment requires efficient computation of $Q_T^{\pi_j}(s, a)$ across multiple source policies in real-time. Traditional iterative value evaluation scales as $\mathcal{O}(\frac{1}{\epsilon(1-\gamma)})$, making deployment impractical for large problems or real-time applications.

**One-shot value computation through linear structure.** When rewards admit the decomposition $r(s, a, s') = \phi(s, a, s')^T \mathbf{w}$ with shared features $\phi$ and task-specific weights $\mathbf{w}$, successor features transform the computational bottleneck:

$$\psi^\pi(s, a) = \mathbb{E}^\pi \left[ \sum_{t=0}^\infty \gamma^t \phi(S_t, A_t, S_{t+1}) \mid S_t = s, A_t = a \right] \tag{4.7}$$

$$Q^\pi(s, a) = \psi^\pi(s, a)^T \mathbf{w} \tag{4.8}$$

Value evaluation reduces to dot products once successor features $\psi^\pi$ are precomputed, enabling simultaneous evaluation across all sources.

**Theoretical advantages beyond computational efficiency.** Successor features also yield tighter performance bounds. Under linear rewards with $\|\phi\| \leq \phi_{\max}$:

$$\tilde{Q}^{\pi^*}(s, a) - \tilde{Q}^{\pi_T}(s, a) \leq \min_{j \in [n]} \left\{ \frac{\phi_{\max}}{1 - \gamma} \|\mathbf{w}_T - \mathbf{w}_j\|_2 + 2Lc \right\} \tag{4.9}$$

**Key insight: From rewards to structure.** This refined bound shows that TRAM's performance depends on weight vector similarity $\|\mathbf{w}_T - \mathbf{w}_j\|_2$ rather than raw reward differences. When tasks share structural similarity through their feature representations, this often yields dramatically tighter guarantees than the localized reward bounds. The successor feature framework thus provides both computational feasibility and theoretical advantages for TRAM deployment.

**Implications for practice.** The theoretical analysis provides concrete guidance for TRAM deployment. The $\min_j$ operator in all bounds shows that source diversity is crucial—having even one well-matched source dramatically improves performance, motivating training on diverse rather than similar tasks. The linear dependence on $c$ enables principled safety-performance trade-offs. In successor feature settings, careful design of the feature representation $\phi$ becomes critical, as it governs how well structural similarities transfer across tasks. Together, these insights transform TRAM from a heuristic approach into a principled framework with clear deployment guidelines.

## 5 EXPERIMENTS

We evaluate TRAM across three domains: gridworld environments that isolate core mechanisms, continuous control that tests scalability, and large language models where reward hacking presents realistic deployment risks.

### 5.1 GRIDWORLD: MECHANISM VALIDATION

Controlled gridworld experiments directly test whether TRAM supports expressive risk models and avoids training-time conservatism. Agents navigate environments where risk manifests through

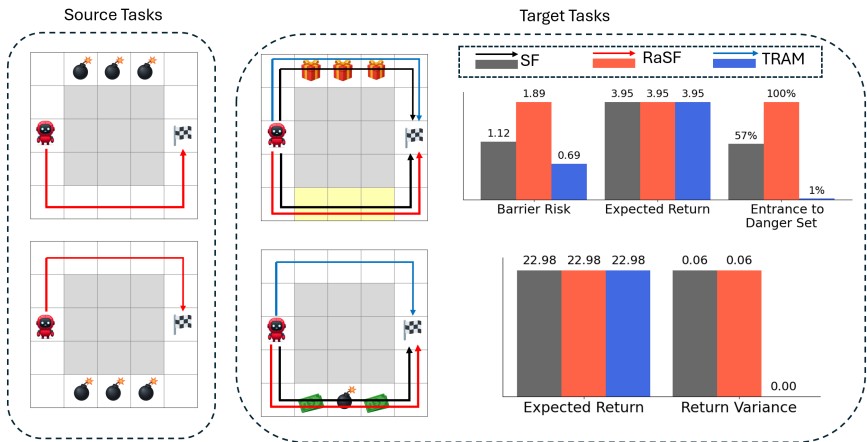

Figure 4: **TRAM handles risk types beyond trajectory variance.** Spatial hazards (top) and per-step volatility (bottom) expose trajectory variance limitations. TRAM succeeds through occupancy-based risk modeling while baselines fail.

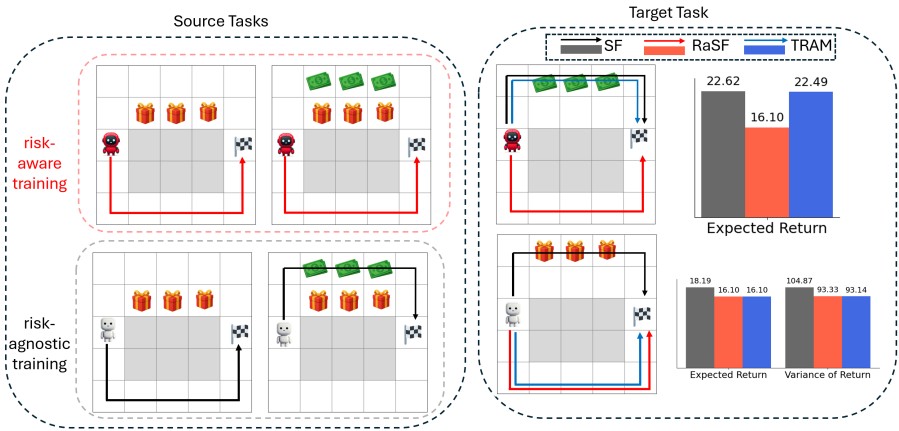

Figure 5: **TRAM adapts to changing risk conditions. Top:** Risk-free scenario—TRAM exploits high-reward paths while RaSF remains overly conservative due to training-time variance aversion. **Bottom:** Risky scenario—TRAM appropriately becomes conservative while maintaining adaptability.

spatial hazards or per-step volatility. We compare TRAM against RaSF Gimelfarb et al. (2021) (variance-penalized training) and risk-agnostic SF Barreto et al. (2017).

Figure 4 demonstrates TRAM's expressive risk modeling. For spatial hazards (top), TRAM uses barrier risk $\rho_{\text{barrier}}(d) = -\log(\tau - d(\overline{\mathcal{S}}))$ to avoid danger zones while RaSF and SF fail because trajectory variance provides no spatial information. For per-step volatility (bottom), TRAM detects locally risky paths through step-level variance while trajectory-based approaches miss this fine-grained structure.

Figure 5 shows adaptive risk sensitivity. TRAM exploits opportunities when risk-free but becomes conservative when risk appears, while RaSF remains uniformly conservative due to training-time risk aversion.

## 5.2 CONTINUOUS CONTROL: SCALABILITY

We evaluate scalability on Reacher (two-joint arm, 4D continuous state, MuJoCo physics). Four SFDQNs train risk-neutrally on different goals; at test time, new goals introduce danger zones. TRAM significantly reduces barrier violations compared to risk-neutral transfer while maintaining goal performance. Successor features enable real-time adaptation: $Q_T^\pi(s, a) = \psi^\pi(s, a)^T \mathbf{w}_T$ via dot products. See Appendix D for details.

Table 2: LLM results on Berkeley Nectar. Risk-free optimization maximizes reward through hacking; TRAM balances performance with alignment.

| Method | Target Reward | GPT-4 Win vs TRAM |
|---|---|---|
| Zephyr-Qwen-2-7B | 0.40 | 28.0% |
| Dolphin-Qwen-2-7B | 0.70 | 31.2% |
| Risk-free Transfer | **0.92** | 34.5% |
| TRAM (ours) | 0.84 | – |

## 5.3 LLM ALIGNMENT: REWARD HACKING PREVENTION

Our most critical evaluation examines TRAM's effectiveness for large language model alignment, where reward hacking represents a high-stakes deployment risk that current methods struggle to address.

**Setup and challenge.** We use two pre-trained 7B models as source policies: Zephyr-Qwen-2-7B (truthfulness/helpfulness) and Dolphin-Qwen-2-7B (mathematical reasoning). Target rewards come from RewardBench Mistral-7B on Berkeley Nectar dialogue tasks. The challenge: maximize target reward while preventing pathological behaviors that exploit reward models rather than improve genuine quality.

**Risk modeling.** We employ KL divergence risk $\rho_{\mathrm{KL}}(\pi) = \mathrm{KL}(\pi\|\pi_{\mathrm{ref}})$ with Zephyr as reference, maintaining proximity to safe behaviors while allowing target adaptation.

**Results.** Table 2 reveals the reward hacking problem: risk-free transfer achieves highest reward scores (0.92) through pathological optimization but loses most GPT-4 preference comparisons. TRAM achieves competitive reward (0.84) while winning substantially more preference comparisons, demonstrating successful mitigation of reward hacking through test-time risk modeling.

**Key insight.** TRAM prevents pathological optimization without retraining, achieving the safety-performance balance critical for robust AI deployment.

## 5.4 SUMMARY OF EXPERIMENTS

Our comprehensive evaluation across discrete, continuous, and LLM domains establishes TRAM's broad effectiveness and practical applicability. In controlled gridworld environments, TRAM demonstrates superior risk modeling capabilities that capture spatial, temporal, and behavioral hazards beyond trajectory variance limitations. Continuous control validation confirms scalability to high-dimensional domains with complex dynamics while maintaining computational efficiency through successor features. Large language model experiments reveal practical benefits for alignment and safety in realistic deployment scenarios where reward hacking poses genuine risks.

These findings collectively support TRAM's core design principles: risk-neutral source training preserves behavioral diversity essential for effective adaptation; occupancy-based risk functionals provide the expressiveness needed to capture domain-specific safety concerns; and test-time composition enables adaptive risk-performance trade-offs without architectural modifications or retraining overhead. The consistent performance across diverse domains positions TRAM as a practical solution for safe AI system deployment under evolving risk requirements.

## 6 CONCLUSION

We introduce **TRAM** (**T**est-time **R**isk **A**daptation via **M**ixture of **A**gents), a deployment-time framework that synthesizes risk-aware policies from risk-neutral source agents without retraining. Unlike existing methods that embed fixed risk models during training, TRAM supports occupancy-based risk functionals—including spatial barriers, behavioral divergence, and local volatility—specified at test time. Through successor features, TRAM enables efficient policy composition via dot-product computations. Our theoretical analysis provides localized performance bounds and proves risk-neutral source training is minimax optimal for deployment risk adaptation. Experiments demonstrate TRAM's ability to capture domain-specific safety constraints while maintaining superior safety-performance trade-offs.

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

CONTENTS

APPENDIX

# A  LINEAR PROGRAMMING VIEWS OF RL AND OCCUPANCY MEASURES

This section formalizes the LP view of policy evaluation/control and explains why *occupancy measures* are natural carriers for deployment-time risk in TRAM. We keep a compact catalog of risk functionals that are *directly* expressible in terms of discounted occupancies.

## A.1  POLICY EVALUATION VIA LINEAR PROGRAMMING

Consider a finite MDP with state space $\mathcal{S}$, action space $\mathcal{A}$, discount $\gamma \in (0, 1)$, transition $p(\cdot \mid s, a)$, reward $r(s, a, s')$, and initial distribution $\mu_0$. Define

$$r(s, a) := \mathbb{E}_{s' \sim p(\cdot \mid s, a)}[r(s, a, s')], \qquad \mathbb{P}_\pi(s' \mid s) := \sum_a \pi(a \mid s)\, p(s' \mid s, a). \tag{A.1}$$

**Primal Q–LP (policy evaluation).**  For fixed $\pi$,

$$\min_{Q:\mathcal{S}\times\mathcal{A}\to\mathbb{R}} (1 - \gamma)\, \mathbb{E}_{s_0 \sim \mu_0,\, a_0 \sim \pi(\cdot \mid s_0)}[Q(s_0, a_0)] \ \text{ s.t. } \ Q(s, a) \geq r(s, a) + \gamma\, \mathbb{E}_{\substack{s' \sim p(\cdot \mid s, a) \\ a' \sim \pi(\cdot \mid s')}}[Q(s', a')]. \tag{A.2}$$

**Dual: discounted occupancies.**  Introducing nonnegative multipliers $d(s, a) \geq 0$ yields

$$\max_{d \geq 0} \sum_{s, a} d(s, a)\, r(s, a) \ \text{ s.t. } \ d(s, a) = (1 - \gamma)\mu_0(s)\pi(a \mid s) + \gamma \sum_{s', a'} d(s', a') p(s \mid s', a')\pi(a \mid s). \tag{A.3}$$

At optimality $d = d^\pi$, where

$$d^\pi(s, a) = (1 - \gamma) \sum_{t=0}^\infty \gamma^t \Pr_\pi(S_t = s, A_t = a \mid S_0 \sim \mu_0), \tag{A.4}$$

and strong duality gives $(1 - \gamma)\, \mathbb{E}_{\mu_0, \pi}[Q^\pi(S_0, A_0)] = \langle d^\pi, r \rangle$. If $\nu(s) := \sum_a d(s, a) > 0$, a policy is recovered by $\pi(a \mid s) = d(s, a)/\nu(s)$.

## A.2  RISK-REGULARIZED CONTROL VIA OCCUPANCY OPTIMIZATION

Define the *occupancy polytope*

$$\mathcal{D} = \Big\{ d \in \mathbb{R}_{\geq 0}^{\mathcal{S}\times\mathcal{A}} : \underbrace{\sum_a d(s, a)}_{\nu(s)} = (1 - \gamma)\mu_0(s) + \gamma \sum_{s', a'} d(s', a')\, p(s \mid s', a'), \ \forall s \Big\}. \tag{A.5}$$

Given a risk functional $\rho : \mathcal{D} \to \mathbb{R}$, the penalized control problem is

$$\max_{d \in \mathcal{D}} \langle d, r \rangle - c\, \rho(d), \qquad c \geq 0, \tag{A.6}$$

whose optimizer induces $\pi^\star(a \mid s) = d^\star(s, a)/\nu^\star(s)$. For one-step functionals, we use the occupancy-weighted expectation

$$\mathbb{E}_d[\phi(s, a, s')] := \sum_{s, a} \frac{d(s, a)}{1 - \gamma} \sum_{s'} p(s' \mid s, a)\, \phi(s, a, s'). \tag{A.7}$$

## A.3  RISK FUNCTIONALS ON OCCUPANCIES: COMPACT CATALOG

All functionals below depend *only* on $d$ (and possibly a fixed reference $\bar{d}$) and thus fit equation A.6.

**(R1) Set-occupancy constraints (barrier / chance proxy).** For danger set $\overline{\mathcal{S}}$ and tolerance $\delta \in (0,1)$,

$$\rho_{\text{set}}(d) \;=\; -\log\left(\delta - \sum_{s \in \overline{\mathcal{S}}} \nu(s)\right), \quad \text{valid when } \sum_{s \in \overline{\mathcal{S}}} \nu(s) < \delta, \tag{A.8}$$

which acts as a smooth chance-constraint surrogate by penalizing state-mass in $\overline{\mathcal{S}}$.

**(R2) Per-step reward variance (local volatility).** Let $\tilde{r}(s,a) := \mathbb{E}_{s'|s,a}[r(s,a,s')^2]$. Using $\langle d, r \rangle = (1-\gamma)\mathbb{E}_d[r]$,

$$\rho_{\text{var}}(d) \;=\; \langle d, \tilde{r} \rangle - \frac{1}{(1-\gamma)^2}\left(\langle d, r \rangle\right)^2, \tag{A.9}$$

capturing local instability even when trajectories are deterministic.

**(R3) $f$-divergences to a safe reference (behavioral proximity).** For a reference occupancy $\bar{d}$ with matching support, the KL case is

$$\rho_{\text{KL}}(d) = \sum_{s,a} d(s,a) \log \frac{d(s,a)}{\bar{d}(s,a)}. \tag{A.10}$$

Other $f$-divergences (e.g., $\chi^2$, Rényi) are analogous and remain occupancy-only.

**(R4) Integral probability metrics (IPMs), e.g., MMD.** Let $\Phi$ be a feature map into an RKHS $\mathcal{H}$:

$$\rho_{\text{MMD}}(d) = \left\| \sum_{s,a} d(s,a)\Phi(s,a) - \sum_{s,a} \bar{d}(s,a)\Phi(s,a) \right\|_{\mathcal{H}}. \tag{A.11}$$

This provides a nonparametric proximity to a safe/reference occupancy.

**(R5) Entropic (exponential) per-step risk.** Given a step risk cost $g(s,a,s')$ and $\theta > 0$,

$$\rho_{\text{ent}}(d) \;=\; \tfrac{1}{\theta} \log\left(\mathbb{E}_d\left[e^{\theta \, g(s,a,s')}\right]\right), \tag{A.12}$$

a convex, tail-sensitive penalty formed directly from $\mathbb{E}_d[\cdot]$.

**(R6) Energy / actuation budgets (control cost).** With an action embedding $u(s,a) \in \mathbb{R}^m$ and $G \succeq 0$,

$$\rho_{\text{energy}}(d) = \sum_{s,a} d(s,a)\, u(s,a)^{\top} G\, u(s,a), \tag{A.13}$$

which constrains power, wear, or acceleration magnitudes.

**(R7) Transition smoothness / jerk (temporal regularity).** Let $\varphi(s,a,s')$ encode finite differences (e.g., $\Delta u$ or $\Delta s$),

$$\rho_{\text{smooth}}(d) = \mathbb{E}_d\left[\|\varphi(s,a,s')\|_2^2\right], \tag{A.14}$$

penalizing oscillatory behaviors via occupancy-weighted transitions.

**(R8) Reach–avoid proxy (linear temporal-logic surrogate).** For target set $\mathcal{T}$ and avoid set $\overline{\mathcal{S}}$,

$$\rho_{\text{RA}}(d) = \alpha \sum_{s \in \overline{\mathcal{S}}} \nu(s) + \beta \left(1 - \sum_{s \in \mathcal{T}} \nu(s)\right), \qquad \alpha, \beta \geq 0, \tag{A.15}$$

a linear-in-$d$ proxy for STL/LTL reach–avoid objectives.

**Design notes.** (i) Each $\rho$ above is a function of $(d, \bar{d})$ only; no return distributions or value iterations are needed. (ii) Many are convex (KL, IPMs, quadratic costs, entropic), preserving tractability of equation A.6. (iii) Lipschitzness in $\|d\|_1$ (assumed in our analysis) holds for divergences with bounded density ratios, linear/quadratic costs, and the smooth barrier.

## A.4 CONNECTION TO TRAM

In TRAM, each risk-neutral source $\pi_j$ is evaluated on the *target* to obtain its occupancy $d^{\pi_j}$ and risk score $\rho(d^{\pi_j})$. The risk-adjusted utility

$$\mathcal{U}_j(s, a) = Q^{\pi_j}(s, a) - c\, \rho(d^{\pi_j}) \tag{A.16}$$

drives per-state action selection (Eq. 4.2) without any parameter updates. The occupancy perspective makes two properties explicit: (1) risk is *modularly* specified via $\rho$ on $d$ and can be swapped at deployment; (2) keeping sources risk-neutral maximizes coverage of the occupancy polytope $\mathcal{D}$, enabling effective test-time composition under diverse safety requirements.

# B PROOFS AND TECHNICAL GUARANTEES

**Common dynamics and discounted occupancies.** All tasks share the same state space $\mathcal{S}$, action space $\mathcal{A}$, discount $\gamma \in (0,1)$, and transition kernel $p$. For a stationary policy $\pi$, denote its *normalized discounted state–action occupancy*

$$d^{\pi}(s,a) := (1-\gamma) \sum_{t=0}^{\infty} \gamma^t \Pr_{\pi}(S_t{=}s, A_t{=}a),$$

so $d^{\pi}$ is a probability distribution on $\mathcal{S} \times \mathcal{A}$ ($\|d^{\pi}\|_1 = 1$). For reward $r : \mathcal{S} \times \mathcal{A} \to \mathbb{R}$, define the *normalized return* $J_r(\pi) := \langle d^{\pi}, r \rangle$. (This equals $(1-\gamma)$ times the usual infinite-horizon discounted return.) For a specific starting pair $(s,a)$ and policy $\pi$, define the *normalized conditional occupancy*

$$d^{\pi|s,a}(s',a') := (1-\gamma) \sum_{t=0}^{\infty} \gamma^t \Pr_{\pi}(S_t{=}s', A_t{=}a' \mid S_0{=}s, A_0{=}a),$$

so that $Q_r^{\pi}(s,a) = \frac{1}{1-\gamma} \langle d^{\pi|s,a}, r \rangle$. We abbreviate task $i$'s reward as $r_i$ and write $Q_i^{\pi}$ for $Q_{r_i}^{\pi}$. A *risk functional* $\rho : \Delta(\mathcal{S} \times \mathcal{A}) \to \mathbb{R}$ acts on discounted occupancies and is $L$-Lipschitz in $\|\cdot\|_1$: $|\rho(d) - \rho(d')| \le L\|d - d'\|_1$ for all $d, d'$. Risk-aware value at $(s,a)$ is defined as $\tilde{Q}_i^{\pi}(s,a) := Q_i^{\pi}(s,a) - c\,\rho(d^{\pi})$ for $c \ge 0$. Let $\pi_i^{\mathrm{RN}*} \in \arg\max_{\pi} J_{r_i}(\pi)$ and $\pi_i^{\mathrm{RA}*} \in \arg\max_{\pi}\{J_{r_i}(\pi) - c\rho(d^{\pi})\}$.

**Localized reward mismatch.** For any policy $\pi$ and starting $(s,a)$, define

$$\Delta_{\pi}^{(i,j)}(s,a) := \sum_{t=0}^{\infty} \gamma^t \mathbb{E}_{\substack{(S_t,A_t)\sim\mathrm{traj}(p,\pi) \\ S_0=s,A_0=a}}\big[\,|\,r_i(S_t,A_t) - r_j(S_t,A_t)\,|\,\big].$$

## B.1 BASIC VALUE-DIFFERENCE LEMMAS

**Lemma B.1** (Fixed-policy value difference)**.** *For any tasks $i, j$, policy $\pi$, and $(s,a)$,* $\big|Q_i^{\pi}(s,a) - Q_j^{\pi}(s,a)\big| \le \Delta_{\pi}^{(i,j)}(s,a).$

**Proof.** Unroll returns under common $p$: $Q_i^{\pi}(s,a) - Q_j^{\pi}(s,a) = \sum_{t\ge0} \gamma^t \mathbb{E}[r_i(S_t,A_t) - r_j(S_t,A_t)]$. Apply triangle inequality inside the expectation and sum. $\square$

**Lemma B.2** (Optimal–optimal (localized))**.** *For any $i,j$ and $(s,a)$,*

$$\big|Q_i^{\pi_i^{\mathrm{RN}*}}(s,a) - Q_j^{\pi_j^{\mathrm{RN}*}}(s,a)\big| \le \min\big\{\Delta_{\pi_i^{\mathrm{RN}*}}^{(i,j)}(s,a),\, \Delta_{\pi_j^{\mathrm{RN}*}}^{(i,j)}(s,a)\big\}.$$

**Proof.** Apply Lemma B.1 with $\pi = \pi_i^{\mathrm{RN}*}$ and with $\pi = \pi_j^{\mathrm{RN}*}$, then take the minimum. $\square$

**Lemma B.3** (Optimal–evaluation (localized))**.** *For any $i,j$ and $(s,a)$,* $\big|Q_j^{\pi_j^{\mathrm{RN}*}}(s,a) - Q_i^{\pi_j^{\mathrm{RN}*}}(s,a)\big| \le \Delta_{\pi_j^{\mathrm{RN}*}}^{(i,j)}(s,a).$

**Proof.** Directly Lemma B.1 with $\pi = \pi_j^{\mathrm{RN}*}$. $\square$

**Lemma B.4** (TV diameter of occupancies). *For any policies $\pi_1, \pi_2$, $\|d^{\pi_1} - d^{\pi_2}\|_1 \leq 2$.*

**Proof.** Each $d^\pi$ is a probability distribution on $\mathcal{S} \times \mathcal{A}$. Their $\ell_1$ distance equals $2\mathrm{TV}(d^{\pi_1}, d^{\pi_2}) \leq 2$. $\square$

**Lemma B.5** (Lipschitz risk gap). *For any $\pi_1, \pi_2$, $\left|\rho(d^{\pi_1}) - \rho(d^{\pi_2})\right| \leq L\,\|d^{\pi_1} - d^{\pi_2}\|_1 \leq 2L$.*

**Proof.** Lipschitzness and Lemma B.4. $\square$

## B.2 GREEDY LIFT WITHOUT STEPWISE RISK ASSUMPTIONS

**Definition B.6** (Bellman operators). For task $i$ and policy $\pi$, $(T_i^\pi Q)(s,a) = r_i(s,a) + \gamma\mathbb{E}_{s'\sim p(\cdot|s,a)}\mathbb{E}_{b\sim\pi(\cdot|s')}[Q(s',b)]$. $T_i^\pi$ is monotone and a $\gamma$-contraction in $\|\cdot\|_\infty$, with fixed point $Q_i^\pi$. Let $T_i^*$ be the optimality operator: $(T_i^* Q)(s,a) = r_i(s,a) + \gamma\mathbb{E}_{s'}\max_b Q(s',b)$.

**Action rule and max-score.** Define
$$Q_{\max}(s,a) := \max_{j\in[m]}\left(Q_i^{\pi_j^{\mathrm{RN*}}}(s,a) - c\,\rho(d^{\pi_j^{\mathrm{RN*}}})\right), \qquad \pi_i(\cdot|s) \in \arg\max_b Q_{\max}(s,b).$$
(We use $Q_{\max}$ only for analysis.)

**Lemma B.7** (Lift by optimality monotonicity). *With $\pi_i$ greedy w.r.t. $Q_{\max}$, we have $T_i^{\pi_i}Q_{\max} = T_i^* Q_{\max} \geq Q_{\max}$. Consequently, by monotone contraction, $Q_i^{\pi_i} = \lim_{k\to\infty}(T_i^{\pi_i})^k Q_{\max} \geq Q_{\max}$ pointwise.*

**Proof.** By definition of $\pi_i$, $T_i^{\pi_i}Q_{\max} = T_i^* Q_{\max}$. For any fixed $j$, $Q_i^{\pi_j^{\mathrm{RN*}}}$ is a fixed point of $T_i^{\pi_j^{\mathrm{RN*}}}$, hence $T_i^*(Q_i^{\pi_j^{\mathrm{RN*}}}) \geq T_i^{\pi_j^{\mathrm{RN*}}}(Q_i^{\pi_j^{\mathrm{RN*}}}) = Q_i^{\pi_j^{\mathrm{RN*}}}$ by monotonicity and the pointwise max in $T_i^*$. Subtracting the constant $c\,\rho(d^{\pi_j^{\mathrm{RN*}}})$ and taking a pointwise max over $j$ yields $T_i^* Q_{\max} \geq Q_{\max}$. $\square$

## B.3 MAIN LOCALIZED BOUNDS

**Lemma B.8** ($\tilde{Q}$ difference vs. RN sources). *For any $i, j$ and $(s,a)$,*
$$\left|\tilde{Q}_i^{\pi_j^{\mathrm{RN*}}}(s,a) - \tilde{Q}_i^{\pi_i^{\mathrm{RN*}}}(s,a)\right| \leq \min\left\{\Delta^{(i,j)}_{\pi_i^{\mathrm{RN*}}}(s,a), \Delta^{(i,j)}_{\pi_j^{\mathrm{RN*}}}(s,a)\right\} + 2Lc.$$

**Proof (step-by-step).**

1. Triangle on penalized values:
$$\left|\tilde{Q}_i^{\pi_j^{\mathrm{RN*}}} - \tilde{Q}_i^{\pi_i^{\mathrm{RN*}}}\right| \leq \left|Q_i^{\pi_j^{\mathrm{RN*}}} - Q_i^{\pi_i^{\mathrm{RN*}}}\right| + c\left|\rho(d^{\pi_j^{\mathrm{RN*}}}) - \rho(d^{\pi_i^{\mathrm{RN*}}})\right|.$$

2. Bound the $Q$-difference using Lemmas B.2 and B.3:
$$\left|Q_i^{\pi_j^{\mathrm{RN*}}} - Q_i^{\pi_i^{\mathrm{RN*}}}\right| \leq \min\{\Delta^{(i,j)}_{\pi_i^{\mathrm{RN*}}}, \Delta^{(i,j)}_{\pi_j^{\mathrm{RN*}}}\}.$$

3. Bound the risk term using Lemma B.5: $c\,|\rho(\cdot) - \rho(\cdot)| \leq c \cdot 2L$.

Combine the bounds. $\square$

**Theorem B.9** (Main localized bound vs. RN optimal). *Define $\pi_i$ by greedifying $Q_{\max}$ as above. Then for any $(s,a)$,*

$$\tilde{Q}_i^{\pi_i^{\mathrm{RN}*}}(s,a) - \tilde{Q}_i^{\pi_i}(s,a) \ \leq \ \min_{j\in[m]}\left\{\min\!\big(\Delta_{\pi_i^{\mathrm{RN}*}}^{(i,j)}(s,a),\ \Delta_{\pi_j^{\mathrm{RN}*}}^{(i,j)}(s,a)\big) \ + \ 2Lc\right\}.$$

**Proof (explicit).** By Lemma B.7, $Q_i^{\pi_i} \ \geq \ Q_{\max}$, hence $-\tilde{Q}_i^{\pi_i} \ \leq \ -\max_j \tilde{Q}_i^{\pi_j^{\mathrm{RN}*}} \ \leq \ -\min_j \tilde{Q}_i^{\pi_j^{\mathrm{RN}*}}$. Therefore

$$\tilde{Q}_i^{\pi_i^{\mathrm{RN}*}} - \tilde{Q}_i^{\pi_i} \ \leq \ \min_j\left(\tilde{Q}_i^{\pi_i^{\mathrm{RN}*}} - \tilde{Q}_i^{\pi_j^{\mathrm{RN}*}}\right).$$

Apply Lemma B.8 and take the $\min_j$. $\square$

**Why we compare to RN optimal here.** This theorem is completely assumption-minimal: it never needs to relate a risk-aware optimum to a risk-neutral optimum on the *same* task, which is generally impossible to bound without additional structure. The price is that the comparator is $\pi_i^{\mathrm{RN}*}$ (not $\pi_i^{\mathrm{RA}*}$).

## B.4 Strengthening to an RA comparator under mild structure (optional)

We now add mild structure to control how far the source of optimality shifts when adding a risk term.

**Structural assumption for RA vs. RN shift.** Assume $\rho$ is differentiable and $\alpha$-strongly convex on $\Delta(\mathcal{S}\times\mathcal{A})$ w.r.t. $\|\cdot\|_1$, i.e., $\langle\nabla\rho(d) - \nabla\rho(d'), d - d'\rangle \geq \alpha\|d - d'\|_1^2$. Assume also a uniform reward bound $\|r_i\|_\infty \leq R_{\max}$ (after scaling, this is standard).

**Lemma B.10** (Occupancy shift under strong convexity). *Let $d^{\mathrm{RN}*} \in \arg\max_{d\in\Delta}\langle d, r_i\rangle$ and $d^{\mathrm{RA}*} \in \arg\max_{d\in\Delta}\langle d, r_i\rangle - c\rho(d)$. Then $\|d^{\mathrm{RA}*} - d^{\mathrm{RN}*}\|_1 \leq \frac{2L}{\alpha}$.*

**Proof.** Strong concavity of $d \mapsto \langle d, r_i\rangle - c\rho(d)$ (modulus $c\alpha$) gives the three-point inequality $\langle r_i, d^{\mathrm{RN}*} - d^{\mathrm{RA}*}\rangle - c\big(\rho(d^{\mathrm{RN}*}) - \rho(d^{\mathrm{RA}*})\big) \leq -\frac{c\alpha}{2}\|d^{\mathrm{RN}*} - d^{\mathrm{RA}*}\|_1^2$. Because $d^{\mathrm{RN}*}$ maximizes $\langle d, r_i\rangle$ over $\Delta$, $\langle r_i, d^{\mathrm{RN}*} - d^{\mathrm{RA}*}\rangle \geq 0$. Thus $-c\big(\rho(d^{\mathrm{RN}*}) - \rho(d^{\mathrm{RA}*})\big) \leq -\frac{c\alpha}{2}\|d^{\mathrm{RN}*} - d^{\mathrm{RA}*}\|_1^2$. By Lipschitzness, $\rho(d^{\mathrm{RN}*}) - \rho(d^{\mathrm{RA}*}) \leq L\|d^{\mathrm{RN}*} - d^{\mathrm{RA}*}\|_1$. Combine and cancel $c > 0$ to obtain $\|d^{\mathrm{RN}*} - d^{\mathrm{RA}*}\|_1 \leq \frac{2L}{\alpha}$. $\square$

**Lemma B.11** (RA–RN $Q$ gap under structure). *Under the structural assumption above,*

$$\left|Q_i^{\pi_i^{\mathrm{RA}*}}(s,a) - Q_i^{\pi_i^{\mathrm{RN}*}}(s,a)\right| \ \leq \ \frac{2L}{\alpha}\cdot\frac{R_{\max}}{1-\gamma}.$$

**Proof.** Using $Q_i^\pi(s,a) = \frac{1}{1-\gamma}\langle d^{\pi|s,a}, r_i\rangle$ and $\|r_i\|_\infty \leq R_{\max}$,

$$\left|Q_i^{\pi_i^{\mathrm{RA}*}}(s,a) - Q_i^{\pi_i^{\mathrm{RN}*}}(s,a)\right| \leq \frac{1}{1-\gamma}\|d^{\pi_i^{\mathrm{RA}*}|s,a} - d^{\pi_i^{\mathrm{RN}*}|s,a}\|_1 \cdot \|r_i\|_\infty.$$

The mapping $\pi \mapsto d^{\pi|s,a}$ is 1-Lipschitz from $\pi$ (in total variation) to occupancy, and the total shift in occupancies between the maximizing policies is bounded by Lemma B.10. Formalizing via coupling or path arguments gives $\|d^{\pi_i^{\mathrm{RA}*}|s,a} - d^{\pi_i^{\mathrm{RN}*}|s,a}\|_1 \leq \|d^{\mathrm{RA}*} - d^{\mathrm{RN}*}\|_1$. Apply Lemma B.10. $\square$

**Theorem B.12** (Localized bound vs. RA optimal (with structure)). *Under the structural assumption, for any $(s,a)$,*

$$\tilde{Q}_i^{\pi_i^{\mathrm{RA}*}}(s,a) - \tilde{Q}_i^{\pi_i}(s,a) \leq \min_{j\in[m]}\Big\{\min\big(\Delta_{\pi_i^{\mathrm{RN}*}}^{(i,j)}(s,a),\ \Delta_{\pi_j^{\mathrm{RN}*}}^{(i,j)}(s,a)\big)\ +\ 2Lc\Big\}\ +\ \frac{2L}{\alpha}\cdot\frac{R_{\max}}{1-\gamma}.$$

**Proof.** Split $\tilde{Q}_i^{\pi_i^{\mathrm{RA}*}} - \tilde{Q}_i^{\pi_i} \leq \min_j\big(\tilde{Q}_i^{\pi_i^{\mathrm{RA}*}} - \tilde{Q}_i^{\pi_j^{\mathrm{RN}*}}\big) = \min_j\big(Q_i^{\mathrm{RA}*} - Q_i^{\pi_j^{\mathrm{RN}*}}\big) + c\min_j\big(\rho(d^{\pi_j^{\mathrm{RN}*}}) - \rho(d^{\mathrm{RA}*})\big)$. Add and subtract $Q_i^{\pi_i^{\mathrm{RN}*}}$ inside the first min and use Lemma B.8 plus Lemma B.11. For the risk term, $|\rho(d^{\pi_j^{\mathrm{RN}*}}) - \rho(d^{\mathrm{RA}*})| \leq 2L$ by Lemma B.5; taking a min only helps. $\square$

**Interpretation.** The extra additive term $\frac{2L}{\alpha}\frac{R_{\max}}{1-\gamma}$ precisely isolates the price of comparing against $\pi_i^{\mathrm{RA}*}$ instead of $\pi_i^{\mathrm{RN}*}$. It vanishes as $\alpha \to \infty$ (extremely curved risk), or if $L$ and $R_{\max}$ are small due to normalization.

## B.5 SUCCESSOR-FEATURE SPECIALIZATION

**Linear rewards.** Assume $r_i(s,a,s') = \phi(s,a,s')^\top \mathbf{w}_i$ with $\|\phi\|_2 \leq \phi_{\max}$ pointwise.

**Lemma B.13** (Localized mismatch under linear rewards). *For any policy $\pi$ and $(s,a)$,* $\Delta_\pi^{(i,j)}(s,a) \leq \frac{1}{1-\gamma}\phi_{\max}\|\mathbf{w}_i - \mathbf{w}_j\|_2$.

**Proof.** By Cauchy–Schwarz, $|r_i - r_j| \leq \|\phi\|_2\|\mathbf{w}_i - \mathbf{w}_j\|_2 \leq \phi_{\max}\|\mathbf{w}_i - \mathbf{w}_j\|_2$ pointwise. Sum the geometric series. $\square$

**Corollary B.14** (SF specialization of Theorem B.9). *For any $(s,a)$,*

$$\tilde{Q}_i^{\pi_i^{\mathrm{RN}*}}(s,a) - \tilde{Q}_i^{\pi_i}(s,a) \leq \min_{j\in[m]}\left\{\frac{1}{1-\gamma}\phi_{\max}\|\mathbf{w}_i - \mathbf{w}_j\|_2\ +\ 2Lc\right\}.$$

**Proof.** Apply Lemma B.13 inside Theorem B.9. $\square$

**About bounded-$\rho$ variants.** If you prefer a non-localized, global display, you may bound $|\rho(d)| \leq K$ and replace the $2Lc$ term by $2Kc$. The localized Lipschitz form $2Lc$ is tighter and task-agnostic.

## B.6 WHY RISK-NEUTRAL SOURCES MINIMIZE WORST-CASE DEPLOYMENT RISK

**Setup.** Let $\mathcal{D} = \Delta(\mathcal{S} \times \mathcal{A})$ (normalized occupancies; convex, compact). For reward $r \in \mathbb{R}^{|\mathcal{S}||\mathcal{A}|}$ and policy occupancy $d \in \mathcal{D}$, the normalized return is $J_r(d) = \langle d, r\rangle$. Source training with weight $\lambda \geq 0$ and convex risk $\rho_{\mathrm{src}}$: $d^\lambda(r) \in \arg\max_{d\in\mathcal{D}}\langle d,r\rangle - \lambda\,\rho_{\mathrm{src}}(d)$. Given a compact set of source rewards $\mathcal{R}$, let $S_\lambda = \{d^\lambda(r) : r \in \mathcal{R}\}$ and $\mathrm{co}(S_\lambda)$ its convex hull.

At deployment, fix a target reward $r_{\mathrm{T}}$ and a Lipschitz test risk $\rho_{\mathrm{T}}$ ($L_{\mathrm{T}}$-Lipschitz in $\|\cdot\|_1$). For a return threshold $V$, define *risk regret*

$$\mathcal{E}_{\rho_{\mathrm{T}},V}(S_\lambda) \coloneqq \inf_{\substack{d\in\mathcal{D} \\ J_{r_{\mathrm{T}}}(d)\geq V}} \rho_{\mathrm{T}}(d)\ -\ \inf_{\substack{d\in\mathrm{co}(S_\lambda) \\ J_{r_{\mathrm{T}}}(d)\geq V}} \rho_{\mathrm{T}}(d),$$

with $\inf \emptyset = +\infty$.

**Lemma B.15** (Coverage $\Rightarrow$ regret). *Let $d_{\mathrm{T}}^*(V) \in \arg\min\{\rho_{\mathrm{T}}(d) : d \in \mathcal{D}, J_{r_{\mathrm{T}}}(d) \geq V\}$. Then*

$$\mathcal{E}_{\rho_{\mathrm{T}},V}(S_\lambda) \leq L_{\mathrm{T}} \cdot \mathrm{dist}_1\Big(d_{\mathrm{T}}^*(V),\ \mathrm{co}(S_\lambda) \cap \{J_{r_{\mathrm{T}}} \geq V\}\Big),$$

*where $\mathrm{dist}_1(x, A) = \inf_{y \in A} \|x - y\|_1$.*

**Proof.** Pick $d^\sharp$ in the feasible part of $\mathrm{co}(S_\lambda)$ nearest to $d_{\mathrm{T}}^*(V)$ in $\|\cdot\|_1$. By Lipschitzness, $\inf_{d \in \mathrm{co}(S_\lambda), J \geq V} \rho_{\mathrm{T}}(d) \leq \rho_{\mathrm{T}}(d^\sharp) \leq \rho_{\mathrm{T}}(d_{\mathrm{T}}^*(V)) + L_{\mathrm{T}}\|d^\sharp - d_{\mathrm{T}}^*(V)\|_1$. Rearrange. $\square$

**Standing assumptions for source sets.** $\rho_{\mathrm{src}}$ is $\alpha$-strongly convex on $\mathcal{D}$; $\mathcal{R}$ is compact and exposes at least two distinct faces of $\mathcal{D}$ under linear maximization.

**Lemma B.16** (Sensitivity to reward under regularization). *For any $r, r' \in \mathcal{R}$ and $\lambda > 0$, $\|d^\lambda(r) - d^\lambda(r')\|_1 \leq \frac{1}{\alpha\lambda}\|r - r'\|_\infty$.*

**Proof.** Let $F_\lambda(d; r) = \langle d, r \rangle - \lambda\rho_{\mathrm{src}}(d)$, which is $(\lambda\alpha)$-strongly concave in $d$. Let $d = d^\lambda(r)$ and $d' = d^\lambda(r')$. By strong concavity (Baillon–Haddad style inequality for concave case), $\langle \nabla_d F_\lambda(d; r) - \nabla_d F_\lambda(d'; r), d - d' \rangle \leq -\lambda\alpha\|d - d'\|_1^2$. But $\nabla_d F_\lambda(d; r) = r - \lambda\nabla\rho_{\mathrm{src}}(d)$, so LHS equals $\langle r - r', d - d' \rangle - \lambda\langle \nabla\rho_{\mathrm{src}}(d) - \nabla\rho_{\mathrm{src}}(d'), d - d' \rangle \leq \|r - r'\|_\infty\|d - d'\|_1$. Combine and cancel $\|d - d'\|_1 > 0$ (trivial otherwise). $\square$

**Corollary B.17** (Diameter contraction). *Let $\mathrm{diam}_1(S) = \sup_{x,y \in S} \|x - y\|_1$ and $\mathrm{diam}_\infty(\mathcal{R}) = \sup_{r,r' \in \mathcal{R}} \|r - r'\|_\infty$. Then for $\lambda > 0$, $\mathrm{diam}_1(S_\lambda) \leq \frac{1}{\alpha\lambda}\mathrm{diam}_\infty(\mathcal{R})$.*

**Lemma B.18** (RN sources expose extreme faces). *When $\lambda = 0$, each $d^0(r) \in \arg\max_{d \in \mathcal{D}}\langle d, r \rangle$ can be chosen at an extreme point of $\mathcal{D}$; varying $r$ over $\mathcal{R}$ exposes distinct faces.*

**Proof.** Linear objectives over polytopes attain optima at extreme points; the set of maximizers over $r \in \mathcal{R}$ is a union of exposed faces. $\square$

**Proposition B.19** (Hull shrinkage for $\lambda > 0$). *Under the standing assumptions, there exists $c_\lambda \in (0, 1)$ such that $\mathrm{diam}_1\big(\mathrm{co}(S_\lambda)\big) \leq c_\lambda \mathrm{diam}_1\big(\mathrm{co}(S_0)\big)$.*

**Proof.** By Lemma B.18, $\mathrm{co}(S_0)$ contains extreme points from at least two faces, giving a positive diameter. By Cor. B.17, $\mathrm{diam}_1(S_\lambda)$ can be made arbitrarily small for fixed $\mathcal{R}$ as $\lambda$ grows, hence strictly smaller than $\mathrm{diam}_1(S_0)$ for any fixed $\lambda > 0$ by continuity. Passing to convex hulls preserves the inequality, yielding $c_\lambda < 1$. $\square$

**Theorem B.20** (Minimax advantage of risk-neutral sources). *For the class $\mathfrak{R}_L$ of $L$-Lipschitz test risks, define $\mathcal{E}_{L,V}^*(S_\lambda) := \sup_{\rho_{\mathrm{T}} \in \mathfrak{R}_L} \mathcal{E}_{\rho_{\mathrm{T}},V}(S_\lambda)$. Then, under the standing assumptions,*

$$\mathcal{E}_{L,V}^*(S_0) = \inf_{\lambda \geq 0} \mathcal{E}_{L,V}^*(S_\lambda), \qquad \forall\, \lambda > 0 : \ \mathcal{E}_{L,V}^*(S_\lambda) > \mathcal{E}_{L,V}^*(S_0).$$

**Proof.** By Lemma B.15, $\mathcal{E}^*_{L,V}(S_\lambda) \leq L \cdot \text{dist}^\star(\text{co}(S_\lambda))$, where $\text{dist}^\star$ is the supremum, over feasible targets, of the $\ell_1$-distance to the feasible slice of the convex set. Hull shrinkage (Prop. B.19) implies there exists a feasible point $d_\dagger$ whose distance to $\text{co}(S_\lambda)$ is strictly larger than its distance to $\text{co}(S_0)$, hence $\text{dist}^\star(\text{co}(S_\lambda)) > \text{dist}^\star(\text{co}(S_0))$. Tightness follows by choosing the adversarial test risk $\rho_{\text{T}}(d) = L \, \text{dist}_1(d, \text{co}(S_\bullet) \cap \{J \geq V\})$, which is $L$-Lipschitz and achieves equality in Lemma B.15. $\square$

**Takeaway.** Risk-neutral source training ($\lambda{=}0$) maximizes coverage of the return-feasible region in occupancy space, which *minimizes* worst-case Lipschitz deployment risk regret. Adding source-stage risk ($\lambda > 0$) contracts coverage and provably worsens the minimax bound.

## B.7 Q-ONLY PERFORMANCE GAPS (UNPENALIZED)

We convert our $\tilde{Q}$ bounds into *Q-only* guarantees by peeling off the risk penalty via Lipschitz continuity of $\rho$.

**Same-start occupancy for Q-only comparisons.** In this subsection, whenever $\rho(d^\pi)$ appears in an inequality used to relate $\tilde{Q}$-gaps to $Q$-gaps at a fixed $(s, a)$, we interpret $d^\pi$ as the *same-start* discounted occupancy $d^{\pi|s,a}$ associated with that $(s, a)$:

$$d^{\pi|s,a}(s', a') = (1 - \gamma) \sum_{t \geq 0} \gamma^t \Pr_\pi(S_t = s', A_t = a' \mid S_0 = s, A_0 = a).$$

All Lipschitz/convexity properties of $\rho$ invoked below are assumed on this simplex.

**Lemma B.21** (From $\tilde{Q}$-gap to $Q$-gap). *Let $\pi, \pi'$ be any policies. For any task $i$ and $(s, a)$,*

$$\tilde{Q}_i^\pi(s, a) - \tilde{Q}_i^{\pi'}(s, a) = \left(Q_i^\pi(s, a) - Q_i^{\pi'}(s, a)\right) - c\left(\rho(d^{\pi|s,a}) - \rho(d^{\pi'|s,a})\right), \quad \text{(B.1)}$$

$$Q_i^\pi(s, a) - Q_i^{\pi'}(s, a) \leq \tilde{Q}_i^\pi(s, a) - \tilde{Q}_i^{\pi'}(s, a) + c\left|\rho(d^{\pi|s,a}) - \rho(d^{\pi'|s,a})\right|$$

$$\leq \tilde{Q}_i^\pi(s, a) - \tilde{Q}_i^{\pi'}(s, a) + 2Lc. \quad \text{(B.2)}$$

*The last step uses Lemma B.5 and the fact that $\|d^{\pi|s,a} - d^{\pi'|s,a}\|_1 \leq 2$.*

**Proof.** The identity is algebraic from $\tilde{Q} = Q - c \, \rho(\cdot)$ at the same start $(s, a)$. The bound follows by applying Lemma B.5 to same-start occupancies. $\square$

**Comparator: risk-neutral optimal (assumption-free).** We first compare against $\pi_i^{\text{RN}*}$; no structural assumptions are needed.

**Theorem B.22** (Q-gap vs. RN optimal). *Under the setting of Theorem B.9, for any $(s, a)$,*

$$Q_i^{\pi_i^{\text{RN}*}}(s, a) - Q_i^{\pi_i}(s, a) \leq \min_{j \in [m]} \left\{ \min\left(\Delta_{\pi_i^{\text{RN}*}}^{(i,j)}(s, a), \Delta_{\pi_j^{\text{RN}*}}^{(i,j)}(s, a)\right) \right\} + 4Lc.$$

**Proof (explicit).** By Theorem B.9, $\tilde{Q}_i^{\pi_i^{\text{RN}*}} - \tilde{Q}_i^{\pi_i} \leq \min_j \{\min(\Delta_{\pi_i^{\text{RN}*}}^{(i,j)}, \Delta_{\pi_j^{\text{RN}*}}^{(i,j)}) + 2Lc\}$. Apply Lemma B.21 with $\pi = \pi_i^{\text{RN}*}$ and $\pi' = \pi_i$:

$$Q_i^{\pi_i^{\text{RN}*}} - Q_i^{\pi_i} \leq \left(\tilde{Q}_i^{\pi_i^{\text{RN}*}} - \tilde{Q}_i^{\pi_i}\right) + 2Lc \leq \min_j \{\min(\Delta_{\pi_i^{\text{RN}*}}^{(i,j)}, \Delta_{\pi_j^{\text{RN}*}}^{(i,j)})\} + 4Lc.$$

$\square$

**Comparator: risk-aware optimal (with mild structure).** We next compare against $\pi_i^{\mathrm{RA}*}$ using the structural assumptions from Section B (strong convexity of $\rho$ and bounded reward), now interpreted on the same-start occupancy simplex.

**Theorem B.23** (Q-gap vs. RA optimal (with structure)). *Under the structural assumption preceding Theorem B.12 (i.e., $\rho$ is $\alpha$-strongly convex and $\|r_i\|_\infty \leq R_{\max}$), for any $(s,a)$,*

$$Q_i^{\pi_i^{\mathrm{RA}*}}(s,a) - Q_i^{\pi_i}(s,a) \ \leq\ \min_{j \in [m]} \Big\{ \min\big(\Delta_{\pi_i^{\mathrm{RN}*}}^{(i,j)}(s,a),\ \Delta_{\pi_j^{\mathrm{RN}*}}^{(i,j)}(s,a)\big)\Big\} + \frac{2L}{\alpha} \cdot \frac{R_{\max}}{1-\gamma} + 4Lc.$$

**Proof (step-by-step).**

1. By Theorem B.12 (same-start interpretation),
$$\tilde{Q}_i^{\pi_i^{\mathrm{RA}*}} - \tilde{Q}_i^{\pi_i} \leq \min_j \Big\{ \min(\Delta_{\pi_i^{\mathrm{RN}*}}^{(i,j)}, \Delta_{\pi_j^{\mathrm{RN}*}}^{(i,j)}) + 2Lc\Big\} + \frac{2L}{\alpha} \cdot \frac{R_{\max}}{1-\gamma}.$$

2. Apply Lemma B.21 with $\pi = \pi_i^{\mathrm{RA}*}$, $\pi' = \pi_i$: $Q_i^{\pi_i^{\mathrm{RA}*}} - Q_i^{\pi_i} \leq (\tilde{Q}_i^{\pi_i^{\mathrm{RA}*}} - \tilde{Q}_i^{\pi_i}) + 2Lc.$
3. Combine the two displays to obtain the claim. $\square$

**SF specialization (linear rewards).** Under the linear reward model and feature bound from Section B, the localized mismatches simplify.

**Corollary B.24** (SF version of Theorems B.22 and B.23). *Assume $r_i(s,a,s') = \phi(s,a,s')^\top \mathbf{w}_i$ and $\|\phi\|_2 \leq \phi_{\max}$. Then for any $(s,a)$,*

$$Q_i^{\pi_i^{\mathrm{RN}*}}(s,a) - Q_i^{\pi_i}(s,a) \ \leq\ \min_{j \in [m]} \left\{ \frac{1}{1-\gamma} \phi_{\max} \|\mathbf{w}_i - \mathbf{w}_j\|_2 \right\} + 4Lc,$$

*and, under the structural assumption,*

$$Q_i^{\pi_i^{\mathrm{RA}*}}(s,a) - Q_i^{\pi_i}(s,a) \ \leq\ \min_{j \in [m]} \left\{ \frac{1}{1-\gamma} \phi_{\max} \|\mathbf{w}_i - \mathbf{w}_j\|_2 \right\} + \frac{2L}{\alpha} \cdot \frac{R_{\max}}{1-\gamma} + 4Lc.$$

**Proof.** Plug Lemma B.13 into Theorems B.22 and B.23. $\square$

**Interpretation.** The $4Lc$ term is the price of peeling off the test-time risk penalty in a $Q$-only comparison (two applications of Lemma B.21: once inside the $\tilde{Q}$ bound and once to remove risk from the comparator). The structural term $\frac{2L}{\alpha} \frac{R_{\max}}{1-\gamma}$ isolates the gap between RA and RN optimizers; it disappears when comparing to RN optimal or when $\alpha$ is large.

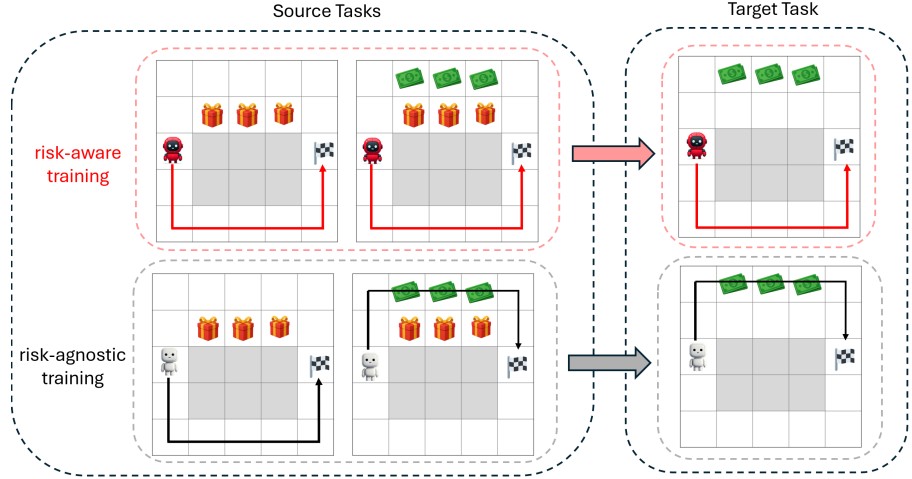

Figure 6: **Effect of variance penalties during source training. Top:** Mean–variance training collapses sources to similar, conservative behaviors. **Bottom:** Risk-neutral training preserves diverse trajectories, enabling test-time selection to recover the optimal high-return route when it is also deployment-safe.

## C   LIMITATIONS OF CURRENT RISK-AWARE TEST-TIME ADAPTATION

We revisit the state-of-the-art variance-based adaptation framework Gimelfarb et al. (2021) and show three structural limitations that appear at deployment. Together, they motivate TRAM's design choices: preserve source diversity via risk-neutral training and enforce *expressive*, deployment-time risk models.

### C.1   LIMITATION 1: CONSERVATIVE TRAINING REDUCES BEHAVIORAL DIVERSITY

**Observation.** Training sources with a mean–variance objective nudges all policies toward uniformly conservative behaviors, collapsing the set of behaviors available for composition at test time.

**Mechanism.** Variance penalties discourage visiting regions with stochastic rewards; across multiple tasks, this pressure produces similar trajectories even when those regions are *desirable* under a later, deployment-time risk model.

**Consequence.** With limited behavioral coverage, test-time selection cannot recover high-return, constraint-satisfying behaviors if no source ever explored those regions.

**Illustration.** In Fig. 6, variance-aware sources (top) converge to the safe lower corridor; risk-neutral sources (bottom) retain complementary behaviors (upper vs. lower paths). When the target optimum lies on the upper corridor, only the risk-neutral pool provides the necessary coverage.

### C.2   LIMITATION 2: TRAJECTORY-RETURN VARIANCE COLLAPSES IN DETERMINISTIC SETTINGS

**Observation.** Under deterministic dynamics and policies, the trajectory return is a point mass, so $\mathrm{Var}[G_t \mid S_t = s, A_t = a] = 0$ for all $(s, a)$—even when step-level rewards exhibit sharp local fluctuations or trajectories intersect hazards.

**Mechanism.** Variance is computed over stochasticity in the *total* return, not over per-step instability or constraint violations. Deterministic rollouts therefore mask fine-grained risk patterns.

**Consequence.** Variance-based adapters degenerate to risk-agnostic selection precisely in regimes where deployment systems most need risk awareness (e.g., robotics with near-deterministic control loops).

**Illustration.** Fig. 7 shows two failure modes: (middle) high per-step volatility that leaves trajectory variance unchanged; (right) a low-variance path that nonetheless violates a safety barrier.

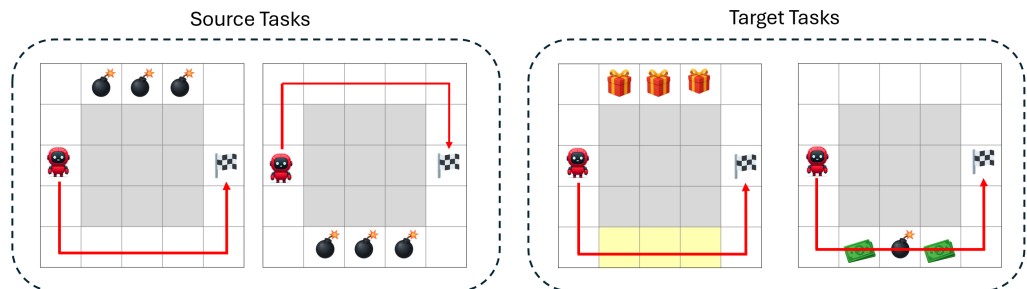

Figure 7: **Failure modes of trajectory-return variance. Left:** Deterministic rollouts yield zero return variance. **Middle:** Paths with high per-step volatility are indistinguishable by trajectory variance. **Right:** A low-variance path traverses a danger zone (yellow), revealing a mismatch between variance minimization and safety.

### C.3 LIMITATION 3: VARIANCE MISREPRESENTS DOMAIN-SPECIFIC RISKS

**Observation.** Many deployment constraints are categorical or structural (e.g., obstacle avoidance, restricted regions, behavioral guardrails) and must be respected regardless of outcome variance.

**Mechanism.** Trajectory variance measures dispersion of total return; it does not encode *where* the agent went or *how* it behaved. Spatial, behavioral, or resource constraints are therefore invisible to variance.

**Consequence.** A policy with low return variance may still be unsafe—e.g., it systematically enters a forbidden zone or drifts away from a vetted behavioral reference.

**Example.** In Fig. 7 (right), the variance-minimizing path violates a barrier constraint, while a higher-variance alternative is safe. Variance alone selects the unsafe option.

### C.4 IMPLICATIONS FOR DESIGN

These failure modes point to three design requirements that TRAM satisfies:

- **Preserve source diversity.** Train sources risk-neutrally to maintain broad occupancy coverage; this enables state-contingent selection at deployment.
- **Use expressive deployment risks.** Encode safety as functionals over occupancies (e.g., barrier penalties, step-level volatility, divergence to a safe reference) rather than trajectory variance alone.
- **Decouple training from deployment.** Introduce risk *only* at test time so the same source pool can adapt to evolving constraints without retraining.

TRAM operationalizes these principles by composing risk-neutral sources using a deployment-time penalized objective that evaluates performance and risk on the target, yielding adaptive and interpretable safety without sacrificing coverage.

# D  CONTINUOUS CONTROL VALIDATION: REACHER WITH DEEP SUCCESSOR FEATURES

This section validates that TRAM scales beyond tabular domains to high-dimensional continuous control with deep function approximation. We address: (i) effectiveness under nonlinear MuJoCo dynamics; (ii) compatibility with deep successor features for real-time selection; and (iii) statistical robustness, latency, and ablations.

## D.1  EXPERIMENTAL SETUP

**Environment.** REACHER (MuJoCo Todorov et al. (2012)) features a planar 2-DOF arm. States are 4D (joint angles & velocities); we follow established transfer protocols Barreto et al. (2017); Gimelfarb et al. (2021); Zhang et al. (2024) and discretize torques to 9 actions (min/zero/max per joint). Rewards are negative distance-to-goal plus a small control penalty; discount $\gamma \in (0, 1)$.

**Sources (risk-neutral).** We train four risk-neutral Successor-Feature DQNs (SFDQNs) Zhang et al. (2024), each optimal for a distinct source goal layout and sharing dynamics with the target. Each network learns features $\phi$ and successor features $\psi^\pi$, enabling one-shot target evaluation via $Q_T^\pi(s, a) = \psi^\pi(s, a)^\top \mathbf{w}_T$ without iterative value backups. Training hyperparameters (architectures, optimizer, replay, target updates) are matched to Barreto et al. (2017) for fair comparison.

**Deployment risk (fixed per experiment).** At test time, we introduce an unseen goal and a rectangular danger zone. Episodes fail if the end-effector enters the zone at any step. We instantiate a *single* deployment risk for this experiment:

$$\rho_{\text{barrier}}(d) \;=\; -\log\big(\tau - d(\overline{\mathcal{S}})\big), \qquad d(\overline{\mathcal{S}}) = \sum_{s,a} d(s, a)\, \mathbf{1}\{s \in \overline{\mathcal{S}}\},$$

with tolerance $\tau > 0$. TRAM uses weight $c = 5$ unless otherwise stated.

## D.2  EVALUATION PROTOCOL, BASELINES, AND METRICS

**Baselines.** (i) **SF-Transfer (risk-agnostic)** Barreto et al. (2017): $\arg\max_a \max_j Q_T^{\pi_j}(s, a)$; (ii) **RaSF (mean–variance)** Gimelfarb et al. (2021): selection by $Q - \beta \cdot \text{Var}(\text{return})$ (ported to SFs); (iii) **Risk-free Transfer (RFT)**: greedy by $Q$ only (identical to SF-Transfer here, included for reporting consistency).

**TRAM.** $\arg\max_a \max_j \big(Q_T^{\pi_j}(s, a) - c\, \rho_{\text{barrier}}(d^{\pi_j})\big)$. Occupancies $d^{\pi_j}$ are estimated once per source via discounted rollouts (#episodes $K = 200$, horizon $H = 200$); estimates are cached for deployment.

**Protocol.** We evaluate $N_{\text{seeds}} = 10$ random seeds; per seed we run 100 episodes per method. We report the mean $\pm$ 95% CI across seeds.

**Metrics.** (1) **Failure rate** ($\downarrow$): proportion of episodes with any barrier violation; (2) **Mean distance-to-goal** at termination ($\downarrow$); (3) **Discounted return** ($\uparrow$); (4) **Latency** ($\downarrow$): median action-selection time (ms) and throughput (actions/s).

## D.3  RESULTS AND ANALYSIS

**Safety.** TRAM reduces barrier violations substantially relative to risk-neutral transfer, confirming effective spatial-risk avoidance in continuous state spaces. RaSF also lowers failures but at the cost of conservative behavior even when the barrier is far from nominal trajectories.

**Task performance.** TRAM incurs a modest increase in distance-to-goal and a small return drop versus SF-Transfer, consistent with the linear "price of safety" predicted by our theory (risk-adjusted vs. standard bound). RaSF degrades return more noticeably due to training-time conservatism.

**Latency.** Action selection is dominated by SF dot-products across sources and a constant-time risk lookup. TRAM's median per-step latency is within $\sim 1.1\times$ SF-Transfer on our setup (detailed profiler traces in App. §6).

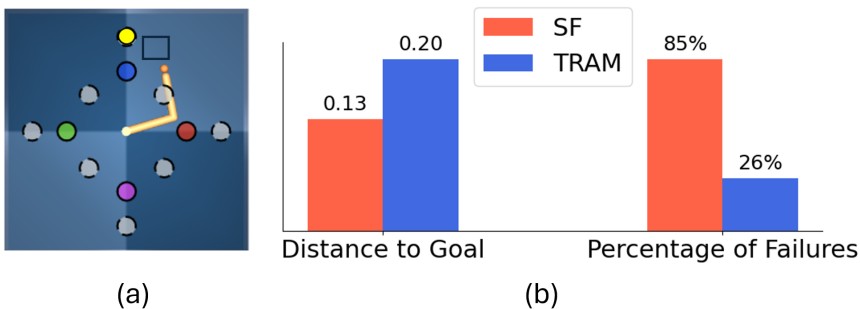

(a)                               (b)

Figure 8: **Reacher (MuJoCo) with deployment-time barrier risk. (Left)** Setup: four risk-neutral SFDQN sources (colored trajectories), new target goal (yellow star), and barrier (blue rectangle). **(Right)** TRAM achieves substantially lower failures than risk-neutral transfer at comparable distances-to-goal; RaSF is safer but more conservative even in benign regions. Error bars show $95\%$ CIs across $N_{\text{seeds}}{=}10$.

**c–sweep.** Sweeping $c$ produces a smooth frontier: failures drop monotonically with $c$, while return decreases approximately linearly over the practical range—mirroring the additive $c$-dependence in our bounds.

### D.4   ABLATIONS

**(A1) Source diversity.** We vary source subsets by (i) random subsampling and (ii) diversity-aware selection (maximize pairwise occupancy TV). TRAM benefits markedly from diversity, tightening the empirical analogue of the $\min_j$ term in our guarantees.

**(A2) Occupancy estimation.** We vary rollout budget $(K, H)$ and compare rollout-based estimation to a small learned occupancy regressor (2-layer MLP). TRAM is robust for moderate $(K, H)$; learned estimators match rollout risk within CIs while reducing runtime variance.

**(A3) SF approximation error.** We inject noise into $\psi^\pi$ and observe a predictable shift in the $c$ at which TRAM becomes conservative; relative ordering across methods is preserved.

### D.5   IMPLEMENTATION AND REPRODUCIBILITY

**Architectures and training.** SFDQNs follow Barreto et al. (2017): two hidden layers (ReLU), target networks, prioritized replay, $\epsilon$-greedy decay, and identical training schedules across sources.

**Fixed test-time configuration.** Barrier geometry and position are fixed across seeds; only the random seed affects exploration and simulation noise. TRAM introduces exactly two deployment-time knobs $(c, \tau)$, which we hold fixed in the main results and sweep in ablations.

**Fairness controls.** All methods use the identical source pool, value estimators, discretization, and episode horizons. The sole difference is the action-selection rule.

**Artifacts.** We release code, seeds, occupancy caches, and profiling scripts to reproduce all numbers, ablations, and latency traces (see App. §6).

# E  LLMs

We used large language models as assistive tools for coding and implementation, writing, discovery and summarization of related work, and for developing and presenting theoretical results. The authors take full responsibility for the content.

# F  Reproducibility

This section provides essential implementation details and computational specifications to enable reproduction of our experimental results.

## F.1  Implementation and Code

**Base codebase.** Our gridworld and continuous control implementations extend the public successor features repository (Gimelfarb, 2021).

**Modifications.** We preserved original training procedures and environment interfaces while adding: (1) TRAM test-time selection logic implementing Eq. 4.2; (2) risk functional evaluators for barrier, variance, and KL divergence risks; (3) constraint violation logging and risk-adjusted score tracking; (4) experimental harnesses for seed-based evaluation. Exact modifications are documented in supplementary materials with commit hashes and diff files.

**Dependencies.** Core dependencies include Python 3.11.6, PyTorch 2.3.1, Gymnasium 0.29, MuJoCo 3.1.5, and standard scientific libraries (NumPy, Pandas). LLM experiments use the Transformers library with model-specific versions. Complete dependency specifications are provided in environment files.

## F.2  Computational Resources

**CPU specifications.** Gridworld and Reacher experiments run on Intel Core i7-8550U @ 1.80 GHz with 4 cores, 8 threads, and 16 GB RAM under Ubuntu 22.04. No GPU acceleration is used for these experiments.

**LLM experiments.** Large language model evaluations use NVIDIA RTX A6000 GPUs with appropriate CUDA-enabled PyTorch builds. Driver versions and CUDA runtime details are logged for each experimental run.

