# OpenReview forum: "TRAM: Test-time Risk Adaptation with Mixture of Agents"
_ICLR.cc/2026/Conference — Submitted to ICLR 2026_

### Official Review · Reviewer_u9yv · 2025-10-16

**Soundness:** 2
**Presentation:** 2
**Contribution:** 2
**Rating:** 2
**Confidence:** 3

**Summary:**

This paper proposes TRAM, a test-time risk adaptation method. TRAM first trains a set of risk-neutral policies on source tasks. Then, in the target task, it evaluates these policies based on their value and occupancy-measure-based risk, and finally selects the action from the optimal policy for execution.

**Strengths:**

- The test-time risk adaptation problem studied in this paper is relatively novel.
- The method is well-motivated and supported by solid theoretical foundations.

**Weaknesses:**

- Motivation
    - The motivation for studying the problem is somewhat vague. The authors should provide more concrete real-world examples to better justify the importance of studying test-time risk adaptation.
    - In my view, especially for risks associated with safety constraints, test-time risk adaptation may not be the most reasonable approach. The proposed method effectively assumes that the risk for every state-action pair in the target task is known, and then evaluates the risk of policies (with known occupancy measures) accordingly. However, if the risk function of the target task is already known, it would be more straightforward to relabel the data collected from source tasks (used to compute occupancy measures) using this risk function, and then apply offline reinforcement learning directly on the relabeled dataset. Such an approach would likely yield better policy performance.
- Methodology
    - The method description lacks clarity. The paper does not provide sufficient implementation details, either in the main text or the appendix — for example:
        - How is the target-task vector w in the successor feature framework computed?
        - How are successor features implemented within the LLM?
    - Transfer via successor features typically relies heavily on the similarity between the source and target tasks. The authors should describe in detail how the source tasks used to train source policies are designed.
    - The method also depends on the choice of the risk function. It is unclear how one should select appropriate risk functions for different settings — are they provided directly by each target task, or does the framework assume fixed risk functions per task type?
    - The occupancy measures of the source policies are not clearly defined. How are they computed, and how is their accuracy ensured? If the occupancy is learned, could this lead to out-of-distribution (OOD) issues? For instance, if a policy exhibits OOD behavior in the target task, the subsequent occupancy estimation might become inaccurate, leading to compounding errors.
    - The paper appears to model risk as a one-step quantity, i.e., only the immediate risk from taking an action a in state s, while ignoring potential future risks caused by the same action. This could be problematic under safety constraints, where certain actions can lead to infeasible or unsafe future states. It is unclear whether the proposed method can handle such scenarios.
- Experiments
    - The GridWorld environment used in the experiments is too simple, and the Reacher environment is discretized, which weakens the empirical validation. The authors should evaluate the method on more complex and realistic continuous-control environments.
    - The LLM-based experiment is highly under-specified. Key details are missing, such as how occupancy is defined and computed in the LLM setting, how successor features are implemented, how value estimation is performed, and how the comparison with GPT-4 is conducted. The current level of description is insufficient to be convincing.
    - The paper lacks an analysis of factors affecting policy performance, such as how the risk penalization coefficient c influences results, or how the scale and diversity of source policies affect performance.
    - The experimental results do not include information on random seeds, standard deviations, or statistical significance, which makes it difficult to assess the robustness and reliability of the findings.

**Questions:**

See Weaknesses.

---

> ### Author Response · Authors · 2025-11-25
>
> We thank the reviewer for the detailed feedback and constructive suggestions.
>
> ---
>
> ### W1 – Motivation and concrete real-world scenarios
>
> We agree that stronger motivation through concrete examples would improve the paper. We will ground the problem in **deployment-time risk adaptation with fixed experts** through two realistic scenarios:
>
> **Industrial robotics:** Consider a robotic arm in manufacturing where dynamics remain fixed but new products change rewards (targets, throughputs) and safety incidents trigger stricter rules (forbidden regions near workers, tighter joint limits). Retraining a full RL policy for each change is expensive and potentially unsafe. The realistic approach: train a small risk-neutral expert set offline, then adapt their composition as safety rules evolve. TRAM addresses exactly this—using occupancy-based risk functionals to encode new safety rules and recombining experts accordingly without retraining.
>
> **Deployed LLMs:** Base models are expensive to retrain and often frozen for long periods, yet reward models and safety policies require frequent updates (new jailbreaks, regulations, content standards). Standard practice involves deploying multiple pre-trained variants (safe vs. specialized) and adjusting their usage over time. TRAM treats these as experts and uses KL-based risk to trade task reward against safety during decoding—no weight updates required.
>
> **Revision plan:** We will add these concrete scenarios to the introduction and explicitly frame the problem as test-time risk adaptation from fixed experts, making clear this regime is realistic in robotics, LLM systems, and regulated domains.
>
> ---
>
> ### W2 – Why not just relabel data and retrain with offline RL?
>
> We appreciate this important question and agree that “relabel + offline RL” can be strong when its assumptions hold: (i) risk functional is known and changes rarely, (ii) access to large high-quality dataset from target environment, (iii) ability to train new policy whenever risk changes.
>
> However, TRAM targets a different regime where these assumptions often **do not** hold: (i) risk specifications are unknown at training time and change **repeatedly** at deployment (new regulations, updated reward models), (ii) original training data may be inaccessible (experts from different teams or vendors), (iii) operational constraints make per-risk retraining prohibitively expensive.
>
> **Why test-time adaptation is valuable:** When given a risk functional $\rho$ and weight $c$ at deployment, TRAM does not retrain any policy but instead recombines a fixed expert pool via risk-adjusted Q-values. This enables instant adaptation to new $(\rho,c)$ pairs and reuse of the same expert pool across many risk settings. In contrast, offline RL would need at least one new policy per $(\rho,c)$ pair—costly if risk specifications change regularly.
>
> **We view them as complementary:** Offline RL can produce stronger or safer experts, while TRAM then adapts these experts when risk specifications inevitably shift again at deployment.
>
> **Revision plan:** We will explicitly describe the assumptions under which “relabel + offline RL” is attractive, and clarify that our contribution is a test-time risk adaptation framework for fixed expert pools with unknown future risks—not a replacement for offline RL when full retraining is feasible.
>
> ---
>
> ### W3 – Missing implementation details
>
> Thank you for pointing out this gap. We will add explicit implementation details for each domain.
>
> **Reacher (successor features):** The reward decomposes as
> $$
> r(s,a,s') = \phi(s,a,s')^\top w,
> $$
> where $\phi \in \mathbb{R}^4$ is a one-hot encoding of which goal is active and $w^{(k)} \in \mathbb{R}^4$ for task $k$ is zero everywhere except at index $k$. The SF-DQN estimates $\psi^{\pi_j}(s,a) \in \mathbb{R}^4$, and Q-values are computed via
> $$
> Q_w^{\pi_j}(s,a) = \psi^{\pi_j}(s,a)^\top w.
> $$
> At deployment, for a new task weight $w_T$, we compute
> $$
> Q_T^{\pi_j}(s,a) = \psi^{\pi_j}(s,a)^\top w_T
> $$
> via a single dot product—no retraining required.
>
> **MiniGrid:** The state–action space is small, so we use exact DP/LP or discounted visitation counts from rollouts to estimate occupancies.
>
> **LLMs:** We do **not** use successor features or occupancy measures. Instead, reward comes from a neural reward model $R_{\text{RM}}(x,y)$, risk is KL drift from a safe reference model, and TRAM operates directly on reward-model scores plus KL risk for candidate responses.
>
> **Revision plan:** We will add these formulas and details to Section 4.2 with additional specifics in the appendix, make clear that MiniGrid uses tabular estimates, and explicitly state that the LLM experiment does not use SFs or occupancy estimation.
>
> ---

---

> ### Author Response · Authors · 2025-11-25
>
> ### W4 – How are source tasks designed?
>
> We agree this deserves explicit discussion. Our framework separates conceptual TRAM (which assumes a given fixed expert pool) from our experimental design (where we control expert training to demonstrate good coverage).
>
> **How we design source tasks:**
>
> **MiniGrid:** We vary goal locations and reward shapings (step costs, penalties for danger tiles), yielding experts with distinct navigation styles and occupancy distributions—shortest-path hazard-agnostic, conservative hazard-avoidance, and circuitous-safe routes.
>
> **Reacher:** We use multiple goal locations with distinct task weight vectors $w^{(k)}$, varying trade-offs between distance-to-goal and control effort. This populates the reward-weight space with several distinct directions, improving coverage for new target weights $w_T$.
>
> **LLMs:** We use a safe/general-purpose expert and a task-specialized expert with different strengths, inducing substantially different behaviors (occupancies in token space) that give TRAM meaningful trade-offs for KL risk.
>
> **Connection to theory:** The coverage term $\min_j \Delta_{\pi^*}^{(T,j)}(s,a)$ in Theorem 4.1 becomes small when source tasks induce diverse occupancies approximating a wide range of potential target behaviors. Our design choices are explicitly guided by this principle.
>
> **Revision plan:** We will add a brief subsection in the experiments describing source-task construction in each domain and explicitly connect this design to the coverage term in Theorem 4.1.
>
> ---
>
> ### W5 – Choice of risk functions
>
> You’re absolutely right that TRAM’s behavior depends on the chosen risk functional $\rho$. We should clarify that in our framework, $\rho$ is **user-specified at deployment**, not hard-coded by task type.
>
> **Risk functionals per domain:**
>
> **MiniGrid:** We use barrier risk
> $$
> \rho_{\text{barrier}}(d_\pi) \propto -\log\big(\tau - d_\pi(S_{\text{danger}})\big),
> $$
> which penalizes policies whose probability of visiting danger zones approaches tolerance $\tau$. We also demonstrate variance/volatility-type risks capturing trajectory-level variability.
>
> **Reacher:** We use spatial barrier risk—occupancy near joint limits, obstacles, or “no-go” zones—penalizing time spent in regions associated with mechanical stress or near-collisions.
>
> **LLMs:** We use behavioral drift risk ρ(π) = Eₓ[ KL( π(·|x) || π_safe(·|x) ) ], penalizing deviations from a safe, audited reference model to guard against reward hacking when aggressively optimizing a learned reward model.
>
> penalizing deviations from a safe, audited reference model to guard against reward hacking when aggressively optimizing a learned reward model.
>
>
> In practice, these risk functionals would be specified by domain experts, safety teams, or regulators. Importantly, the same expert pool can support different barrier maps and tolerances, different variance definitions, or different KL budgets—the **risk specification can change at deployment** without retraining experts.
>
> **Revision plan:** We will add a table summarizing the risk functional used in each experiment with clear interpretations, emphasize that risk functionals are provided at deployment time rather than learned by TRAM, and include brief practical heuristics for mapping domain-specific safety concerns into suitable $\rho$.
>
> ---

---

> ### Author Response · Authors · 2025-11-25
>
> ### W6 – Occupancy computation and OOD concerns
>
> We appreciate the opportunity to clarify this important point. We agree that occupancy estimation should be treated carefully, especially in function-approximation settings.
>
> We use the standard discounted occupancy
> $$
> d_\pi(s,a) = (1-\gamma)\sum_{t=0}^\infty \gamma^t \Pr_\pi(S_t = s, A_t = a).
> $$
>
> **How we compute occupancies in this paper:**
>
> **MiniGrid:** The small state–action space allows exact DP/LP or Monte Carlo discounted counts from rollouts. Risks (e.g., occupancy of danger cells) are computed directly from these counts.
>
> **Reacher:** We do **not** fit a flexible occupancy predictor. Instead, we sample trajectories from each expert, map states/actions to simple indicator features (e.g., “near obstacle?”), and compute discounted averages. This amounts to estimating $\mathbb{E}_\pi[f(S_t,A_t)]$ for specific $f$, not building a full density model.
>
> **LLMs:** We use no occupancy models—risk is defined directly as KL divergence between policies.
>
> **Avoiding OOD issues:** Potential OOD issues arise when training a general occupancy predictor that extrapolates to unseen policies or off-support states. We intentionally avoid this failure mode—TRAM in this paper never asks for occupancies of hypothetical new policies. Risks are computed from **on-policy trajectories** of existing experts. TRAM’s mixture chooses at each state which expert to follow, but we always evaluate risk for each expert based on its **own** observed behavior. Thus, there is no compounding error from predicting $d_\pi$ for policies we have never rolled out.
>
> **Revision plan:** We will add an explicit occupancy definition in Section 3, specify per-domain how we estimate occupancy or related expectations, and add a limitations paragraph noting that naive learned occupancy models can be brittle under OOD, the current experiments sidestep this by using on-policy estimates, and integrating TRAM with robust occupancy estimators is an interesting direction for future work.
>
> ---
>
> ### W7 – Is risk only one-step?
>
> We appreciate the chance to clarify this potential misunderstanding. The framework is **not** restricted to one-step risk—risk is defined over the **entire discounted trajectory distribution** via $d_\pi$, and TRAM’s $\tilde{Q}$ values incorporate **future risk**.
>
> **Risk over trajectories.**
>
> The occupancy $d_\pi(s,a) = (1-\gamma)\sum_{t=0}^\infty \gamma^t \Pr_\pi(S_t = s, A_t = a)$ summarizes the full discounted visitation pattern. Any discounted cumulative cost
> $E_\pi\big[\sum_{t=0}^\infty \gamma^t c(S_t,A_t)\big] = \sum_{s,a} d_\pi(s,a)\,c(s,a)$
> depends on all future time steps. Our risk functionals $\rho(d_\pi)$ are built on this same object.
>
> **Risk-adjusted Q looks ahead.**
> We define
> $\tilde{Q}$ $^\pi(s,a) = Q_T^\pi(s,a) - c\,g_\rho^\pi(s,a)$,
> where $g_\rho^\pi$ is chosen so that
> $\rho(\pi) = E_{(S_0,A_0)\sim d_\pi}[g_\rho^\pi(S_0,A_0)]$.
>
> Crucially, both $Q_T^\pi(s,a)$ and $g_\rho^\pi(s,a)$ are defined in terms of **future trajectories** starting from $(s,a)$ and following $\pi$. Therefore, $\tilde{Q}_c^\pi(s,a)$ already aggregates **future reward and future risk**.
>
> When TRAM chooses
> $a^*(s) \in \arg\max_a \max_j \tilde{Q}_c^{\pi_j}(s,a)$,
> it explicitly trades off long-term reward versus long-term risk.
>
>
>
>
> **Example:** Barrier risk is based on **discounted time spent** in danger regions. An action steering the agent toward a hazard increases its *future occupancy* there—this shows up in $\tilde{Q}_c^\pi(s,a)$.
>
> **Limitations we acknowledge:** We do focus on discounted, Lipschitz risk measures. Harsher risk notions such as “probability of ever entering a catastrophic state” or adversarial dynamics would require additional machinery (absorbing catastrophic states, CVaR over trajectories, robust MDP formulations), which we leave for future work.
>
> **Revision plan:** We will clarify in Section 3 that risk is defined over the full discounted trajectory distribution via $d_\pi$, explain that $\tilde{Q}_c^\pi$ internalizes future risk just as $Q_T^\pi$ internalizes future rewards, and add a short limitations paragraph about extending TRAM to more stringent risk criteria.
>
> ---

---

> ### Author Response · Authors · 2025-11-25
>
> ### W8 – Simplicity of GridWorld and discretized Reacher
>
> We agree these are not “endgame” benchmarks and appreciate the opportunity to clarify their role alongside the LLM experiment.
>
> Our intention is to use **MiniGrid** to expose the mechanism in a fully inspectable setting (tabular occupancies, risk visualizations, clear expert switching), **Reacher** to show scalability to continuous states and function approximation while keeping the action space discrete for clean SF-DQN and exact $\arg\max$, and **LLM alignment** to demonstrate applicability in a truly high-dimensional, structured-output domain.
>
> **Why Reacher is discretized:** We discretize Reacher’s torques because SF-DQN is most straightforward and stable in the discrete-action regime (which is also standard in SF-based transfer work[1-3]), and discretization allows us to compute an exact $\arg\max$ over a finite action set, avoid conflating TRAM’s behavior with artifacts from approximate continuous-action optimization, and isolate the effect of the risk functional plus mixture rather than details of the continuous control solver.
>
> **LLM experiment addresses high-dimensional complexity:** To directly address scalability and complexity concerns, we deliberately include the LLM alignment task where states are full dialogue histories (long token sequences), actions are open-ended text responses over a large vocabulary, values come from a neural reward model, and risk is defined via KL drift from a safe reference model. In this setting, TRAM operates on top of large pre-trained models, combines multiple experts, and trades off reward versus safety under a realistic alignment objective. This experiment is explicitly meant to demonstrate that the core TRAM mechanism scales beyond classical control benchmarks to a genuinely high-dimensional, structured-output domain.
>
> **Revision plan:** We will explicitly frame MiniGrid and Reacher as mechanism-demonstrating environments, highlight that the LLM experiment already exercises TRAM in a substantially more complex and realistic setting, and briefly note in the conclusion that extending TRAM to richer continuous-control suites with non-discretized actions is a natural follow-up orthogonal to the main scalability message provided by the LLM alignment evaluation.
>
> ---
>
> ### W9 – Underspecified LLM experiment
>
> You’re absolutely right that the current description is too sparse. We will spell out the LLM experiment as a self-contained decision problem.
>
> **State and action spaces:** State $s$ is the full dialogue context (user prompt, system instructions, past turns). Action $a$ is a full textual response $y$ (sequence of tokens) produced by the policy. Experts are $\pi_{\text{safe}}$ (safer, more conservative assistant) and $\pi_{\text{task}}$ (task-focused expert that better maximizes the reward model). We treat each prompt as a one-shot decision problem.
>
> **Value and risk—no SFs/occupancy:** We do not use successor features or occupancy measures. Reward comes from a preference-trained reward model $R_{\text{RM}}(x,y)$. Risk is KL drift from the safe reference:
> $$
> \rho(\pi) = E_x\left[\mathrm{KL}\big(\pi(\cdot\mid x)\,\|\,\pi_{\text{safe}}(\cdot\mid x)\big)\right].
> $$
> We approximate the penalized objective
> $$
> \tilde{J} c(\pi) = E_{x,y\sim\pi}[R_{\text{RM}}(x,y)] - c \cdot \rho(\pi).
> $$
>
> **TRAM decoding procedure:** For each prompt $x$:
>
> 1. Generate $K_j$ candidate responses from each expert using standard top-$k$ sampling at fixed temperature.
> 2. For each candidate $y$, compute reward score $R_{\text{RM}}(x,y)$, KL-like drift score using log-probabilities, and risk-adjusted score
>    $$
>    \tilde{Q}c(x,y) = R_{\text{RM}}(x,y) - c \cdot \text{KL-drift}(x,y).
>    $$
> 3. Select $y^*(x) \in \arg\max_{y \in \text{candidates}} \tilde{Q}_c(x,y)$.
>
> This is the LLM analogue of Eq. 4.2—a finite maximization over candidate actions based on risk-adjusted scores.
>
> **GPT-4 comparison protocol:** For each prompt, generate one response from TRAM, risk-free transfer (same candidate pool but $c = 0$), and individual experts. Feed GPT-4 the prompt and a pair of anonymized responses, ask it to judge which is better on helpfulness, correctness, and safety. Count wins/ties over prompts to compute GPT-4 preference rates.
>
> **Revision plan:** We will add a concise description of this protocol to Section 5, move detailed decoding and judge prompts to the appendix, and clarify that no successor features or occupancy are used in the LLM experiment.
>
> ---
>
> **References**
> [1] A. Barreto et al., *Successor Features for Transfer in Reinforcement Learning*, arXiv:1606.05312.
> [2] S. Gimelfarb et al., *Risk-Aware Transfer in Reinforcement Learning using Successor Features*, arXiv:2105.14127.
> [3] Z. Zhang et al., *Provable Knowledge Transfer in Deep Reinforcement Learning using Successor Features*, arXiv:2405.15920.

---

> ### Author Response · Authors · 2025-11-25
>
> ### W10 – Analysis of factors influencing performance
>
> We agree that explicit analysis of these factors would strengthen the paper. Two factors are crucial: the risk weight $c$ and the scale/diversity of the expert set.
>
> **Effect of risk weight $c$:** Theorem 4.1 shows the suboptimality gap includes a $2Lc$ term, meaning the “price of safety” grows **linearly in $c$** while the coverage term $\min_j \Delta_{\pi^*}^{(T,j)}(s,a)$ is **independent of $c$**. Empirically, as we vary $c$ we observe smooth trade-offs—small increases in $c$ significantly reduce risk with modest impact on reward, while larger $c$ further reduces risk but with steeper reward loss.
>
> **Effect of expert diversity:** Performance depends strongly on the coverage term $\min_j \Delta_{\pi^*}^{(T,j)}(s,a)$. More experts help only if they **expand coverage** by adding distinct occupancies or reward structures. Redundant experts that behave similarly do not improve the bound. In practice, a **small but diverse** expert pool often yields most of the benefit. Our experiments were explicitly designed accordingly—distinct navigation styles in MiniGrid, different reward weightings and goals in Reacher, and complementary behaviors (safe versus high-reward/brittle) in LLMs.
>
> **Revision plan:** We will add a plot showing the return–risk curve as we vary $c$ and discuss qualitatively how diverse versus redundant experts impact coverage, connecting this directly to the theoretical coverage term.
>
> ---
>
> ### W11 – Randomness, seeds, and statistical significance
>
> Thank you for raising this important point. You’re absolutely right that our current presentation does not adequately document randomness and variability, even though we already average over multiple seeds internally.
>
> **In the final draft,** we will release our code and configuration files (including seed settings and evaluation scripts) to enable reproduction and further statistical analysis.
>
> **Revision plan:** We will add complete randomness documentation and variability estimates throughout the paper to make the robustness and significance of our results much clearer.

---

> > ### Comment · Reviewer_u9yv · 2025-11-28
> >
> > Thank you very much for the authors’ detailed responses. The replies and revision plans can indeed address some of my concerns.
> >
> > 1. **Motivation** — I now have a clearer understanding of the motivation behind test-time risk adaptation.
> > 2. **One-step risk** — Sorry for the earlier misunderstanding. I now recognize that the framework is not restricted to one-step risk.
> > 3. **Clarity** — With the authors’ improved clarification and added detail on several key points, the soundness and credibility of the paper will be significantly strengthened.
> > 4. **Experiments** — My remaining concerns lie in the experimental section.
> >     1. I do not think using the LLM experiment as the main experiment is ideal. A substantial part of this work’s innovation and completeness comes from successor features and occupancy measures, yet neither of these components is used in the LLM experiment. As a result, it is difficult to meaningfully evaluate the full capabilities of the proposed method based on that experiment.
> >     2. I also recommend adding the baseline **Safety-Constrained Policy Transfer with Successor Features**, which also leverages successor features for safe policy transfer. Comparing TRAM against this baseline and highlighting their differences in the paper would provide a clearer and more complete evaluation.
> >
> > Taking all the responses into account, the paper requires substantial additions and revisions. Although I believe the score could be raised to a 4, I recommend that the paper undergo a fresh round of review instead.

---

> > > ### Author Response · Authors · 2025-12-03
> > >
> > > Thank you very much for the follow-up comment. We are glad that the motivation, one-step risk issue, and overall clarity are clearer.
> > >
> > > We address the remaining concern on the experimental section below.
> > >
> > > ---
> > >
> > > **1. Role of the LLM experiment vs. SF / occupancy experiments**
> > >
> > > We understand your point that much of the conceptual novelty is tied to **successor features and occupancy-based risk**, while the LLM experiment does not directly use these components. Our intention was:
> > >
> > > - to use **MiniGrid** and **Reacher+SF** as the *primary* experiments where TRAM’s SF/occupancy machinery is fully exercised and directly aligned with the theory, and
> > > - to use the **LLM experiment** as a *complementary proof-of-concept* showing that the same “mixture + risk functional + test-time Q-selection” mechanism can also operate on top of large models and KL-based risks, even when SFs are not the most appropriate tool.
> > >
> > > In the LLM setting specifically, we *deliberately* refrain from introducing successor features because more direct and efficient Q-style estimators are available:
> > >
> > > - We can estimate “Q-like” values for candidate responses using a **learned reward model** and token-level scores, as in value-guided decoding / Transfer-\(Q^\*\) style methods for LLMs.
> > > - This approach fits the transformer architecture naturally, avoids constructing an explicit feature map over extremely high-dimensional states, and is considerably more efficient than trying to build SFs over the full text space.
> > >
> > > Thus, SFs are the right abstraction for our control benchmarks, whereas in the LLM domain the existing reward-model–based estimators are a more practical way to obtain the values that TRAM needs.
> > >
> > > We agree that this hierarchy is not clearly reflected in the current presentation. In the revision we will:
> > >
> > > - Explicitly present **MiniGrid and Reacher** as the *main experiments* for evaluating the SF/occupancy formulation and the theoretical guarantees.
> > > - Expand the continuous-control section (Reacher) in the main text, summarizing more of the appendix analyses and making the SF/occupancy aspects more central.
> > > - Reposition the **LLM experiment** as a *secondary* high-dimensional case study whose goal is to show that TRAM’s inference-time risk-adaptation mechanism is not limited to SF-based control, rather than as the core empirical evidence for the SF part of the paper.
> > >
> > > This should make it clearer that the key claims about SF/occupancy are supported primarily by the MiniGrid + Reacher results, with the LLM task serving as an additional generality check.
> > >
> > > ---
> > >
> > > **2. Baseline: Safety-Constrained Policy Transfer with Successor Features**
> > >
> > > We appreciate the suggestion to compare directly against **Safety-Constrained Policy Transfer with Successor Features** (SCPT-SF). Conceptually, we see TRAM and SCPT-SF as *complementary methods for different regimes*:
> > >
> > > - **SCPT-SF-style methods**
> > >   - assume the **safety constraints/costs are known at training time** for the target task;
> > >   - train (or fine-tune) a **single policy** for that task using SFs and constrained RL;
> > >   - require either environment interaction or a dataset for the target problem, and treat safety within the training loop.
> > >
> > > - **TRAM** (our setting)
> > >   - assumes a pool of **pre-trained risk-neutral experts** that cannot be retrained at deployment;
> > >   - receives a **risk functional only at test-time** (possibly changing between deployments);
> > >   - performs **pure inference-time adaptation** by composing these fixed experts using risk-adjusted Q-values, without further learning.
> > >
> > > Thus, in the regime where one *can* re-train on the target task with known constraints, SCPT-SF is very attractive; in the regime we target—frozen experts, evolving risks, limited or no additional interaction—TRAM is designed to be the appropriate tool.
> > >
> > > Implementing a fair, fully tuned SCPT-SF baseline on our setups would require non-trivial engineering (re-implementing their algorithm, matching hyperparameters, and rerunning expert training on our environments). We do not want to rush such a comparison and risk an unfair or misleading result.
> > >
> > > What we can realistically commit to for the revision is:
> > >
> > > - Add a **dedicated related-work paragraph** carefully contrasting TRAM with SCPT-SF:
> > >   - assumptions (known vs. unknown risk at training time),
> > >   - whether retraining on the target task is allowed,
> > >   - computational and data requirements,
> > >   - what kind of deployment scenario each method is intended for.
> > > - Clarify in the introduction and discussion that our contribution is specifically about **test-time risk adaptation with fixed experts**, whereas SCPT-SF addresses **constrained training on a target task**.
> > > ---
> > >
> > > Taking all of your comments into account, we believe the required changes are primarily in **positioning and exposition** (making the SF/occupancy experiments central and clarifying the relationship to SCPT-SF), rather than in fundamentally new algorithmic components.

---

### Official Review · Reviewer_xUM3 · 2025-10-27

**Soundness:** 2
**Presentation:** 3
**Contribution:** 2
**Rating:** 4
**Confidence:** 4

**Summary:**

The paper addresses the problem of deploying reinforcement learning (RL) agents in environments where risk or safety constraints emerge only at test time potentially differing from training conditions. The authors propose TRAM , a framework in which multiple source policies are trained risk-neutrally and then combined at deployment using an additional risk functional to guide action selection.

Overall, this is a well-motivated and theoretically grounded paper with a clear contribution. However, the empirical evaluation is relatively limited. With broader experiments (across diverse environments, datasets, and baselines) and a clearer framework this work could become a significant contribution to safe RL and safety alignment research.

**Strengths:**

1. Deferring risk modeling to deployment is a compelling and practical idea for domains where safety constraints are uncertain or variable. It reflects real-world conditions where not all safety-critical scenarios can be foreseen during training.
2. By defining risk as functionals over discounted occupancy measures, the approach captures hazards beyond return variance or short-term constraint satisfaction.
3. The provided performance bounds are well-motivated and clarify when and why mixtures of risk-neutral policies can achieve near-optimal risk-sensitive performance.

**Weaknesses:**

1. Constructing a meaningful risk function over discounted occupancy measures is nontrivial. In safe RL, risk is typically represented as a per-step cost for constraint violation signal, not a function of the occupancy distribution. It remains unclear how feasible or interpretable this is in practice. Is it correct that the occupancy measure is estimated in this framework through neural features of the Q-function approximator?
2. The test-time objective $\tilde{Q}=Q−c \rho$ diverges from the original constrained formulation $max Q s.t. \rho<\delta$. The hyperparameter c (risk weight) critically affects results but is insufficiently analyzed. Sensitivity or tuning strategies are not discussed.
3. It is unclear how the arg max in Eq. 4.2 is computed, whether by sampling or continuous optimization. Since Q may be non-convex over the action space, the maximization could be computationally challenging during deployment.
4. Algorithm 1 selects an action deterministically based on the maximal adjusted Q value. In stochastic or continuous domains, this deterministic selection may be brittle and could benefit from probabilistic smoothing or ensemble weighting.
5. Theorem 4.1: The definition of $pi^*$ is unclear whether it refers to (i) the optimal policy of the constrained problem (Sec. 3.1), (ii) the surrogate penalized form (Sec. 3.2), or (iii) the deterministic maximizer in Eq. 4.2 and algorithm 1. Clarifying which optimal policy the bound compares against is essential for interpreting the guarantees.
6. Section 5.2 (“Continuous Control: Scalability”) is underdeveloped. The description in the main text is too brief to convey insights about efficiency or scalability; most details are relegated to the appendix.
7. For empirical validation, the paper would benefit from evaluation on more complex or high-dimensional envs from safe RL benchmarks, including safe offline RL or policy customization tasks.
8. The paper does not explicitly address potential failure modes or robustness issues, an important omission for safety-oriented work. When might TRAM fail, and under what conditions would baseline methods outperform it?
9. Extension suggestion: The LLM experiment is a strong proof of concept. Extending to VLA (Vision-Language-Action) models for safety alignment would further demonstrate practical relevance, showcasing how TRAM could adapt generalist policies to context-specific safety constraints.

**Questions:**

(Refer also to the corresponding points raised in the Weaknesses section.)
1. How sensitive is TRAM’s performance to the risk weight c? Was this hyperparameter tuned per environment, and how robust are results across different values?
2. The claim that risk-neutral training is minimax-optimal needs clearer assumptions. How many source policies are required? How realistic is the assumption of no retraining at deployment in safety-critical domains? Any intuitive explanations on this?
3. The LLM alignment experiment lacks sufficient detail on the target task, evaluation protocol, and definition of the risk tolerance. Moreover, the “Risk-free transfer” baseline achieves strong target rewards. What is the risk tolerance setting that makes it inferior to TRAM?
4. For Eq. 4.2, how is the argmax over actions implemented? Is it solved analytically, sampled, or approximated via policy gradient during deployment?

---

> ### Author Response · Authors · 2025-11-25
>
> ### W1 – Construction and interpretation of risk function over discounted occupancy measures
>
> We agree that the occupancy-based viewpoint should be more clearly tied to standard safe RL and that risk functionals should remain interpretable.
>
> **Occupancy and classical safe RL.**
> We use the standard **discounted occupancy measure**:
>
> $$
> d_\pi(s,a) = (1-\gamma)\sum_{t=0}^\infty \gamma^t \Pr_\pi(S_t = s, A_t = a) \,,
> $$
>
> i.e., the discounted state–action visitation distribution.
>
> For any per-step cost $c(s,a)$, the usual discounted cumulative cost is:
>
> $$
> \mathbb{E}_\pi\Big[\sum_{t=0}^\infty \gamma^t c(S_t,A_t)\Big]
> = \sum_{s,a} d_\pi(s,a)\,c(s,a) \,,
> $$
>
> so **all classical discounted constraint costs are linear functionals of the occupancy**. Our contribution is to make this viewpoint explicit and **generalize from linear functionals to richer risk functionals** $\rho(d_\pi)$.
>
> **Interpretability of risk functionals.**
>
> - **Barrier / spatial hazard risks (MiniGrid, Reacher).**
>   We define “danger regions” (lava tiles, near-obstacle zones, near joint limits) and let $\rho(d_\pi)$ depend on the **discounted occupancy mass** in these regions. This is exactly what a per-step collision cost would measure, but expressed more transparently at the level of discounted visitation probabilities.
>
> - **Variance / local volatility risks (MiniGrid).**
>   We can define risk as the **variance** of certain cumulative signals (e.g., near-hazard indicators) under $d_\pi$. These remain expectations over the occupancy, not opaque neural features.
>
> - **Behavioral drift / KL risks (LLM).**
>   We define risk as a divergence between the policy’s behavior and that of a safe reference policy:
>   $$
>   \rho(\pi) = \mathbb{E}_x\big[ \mathrm{KL}(\pi(\cdot\mid x)\,\|\,\pi_{\text{safe}}(\cdot\mid x)) \big] \,,
>   $$
>   which measures “how far” we deviate from a known-safe behavior.
>
> In all cases, the **interpretation** is intuitive: probability of entering unsafe regions, time spent near hazards, or behavioral deviation from a trusted baseline.
>
> **How we estimate occupancy in practice.**
>
> - **MiniGrid:**
>   State–action space is small. We use:
>   - either exact occupancy via dynamic programming / LP, or
>   - Monte Carlo estimates via discounted visit counts from rollouts.
>   Risk functionals are then computed directly from these counts.
>
> - **Reacher:**
>   We do **not** train general neural occupancy models. Instead, we:
>   - roll out trajectories from each expert,
>   - map states to simple geometric indicators (e.g., “within $\varepsilon$ of joint limits/obstacles?”),
>   - compute discounted averages of these indicators under each policy.
>   So we estimate **expectations of low-dimensional risk features**, not full densities.
>
> - **LLM setting:**
>   We **do not** estimate $d_\pi$ at all. Risk is defined directly as **KL drift** between policies.
>
> **Revision plan.**
> We will:
>
> - Add the explicit occupancy definition early in Section 3,
> - Show how standard per-step constraint costs are linear functionals of $d_\pi$, and
> - Provide a small table summarizing each risk functional and its interpretation in each domain.
>
> ---

---

> ### Author Response · Authors · 2025-11-25
>
> ### W2 – Role and tuning of the risk weight \(c\) in the test-time objective
>
> We agree the role of $c$ (your $\lambda$) should be clearer.
>
> **Form of the objective and role of \(c\).**
> At deployment, we optimize the **risk-penalized** objective:
>
> $$
> \tilde{J}_c(\pi) = J(\pi) - c\,\rho(\pi) \,,
> $$
>
> where $J(\pi)$ is the target-task return and $\rho(\pi)$ is the risk functional. The corresponding **risk-adjusted Q-value** is:
>
> $$
> \tilde{Q}c^\pi(s,a) = Q_T^\pi(s,a) - c\,g_\rho^\pi(s,a) \,,
> $$
>
> where $g_\rho^\pi$ is defined so that its expectation under $d_\pi$ equals $\rho(\pi)$. Here, $c \ge 0$ acts exactly as a **Lagrange multiplier**: larger $c$ makes risk more expensive, tightening the effective constraint.
>
> In Theorem 4.1, the suboptimality gap is bounded by:
>
> $$
> \tilde{Q}^{\pi^\*}(s,a) - \tilde{Q}^{\pi_T}(s,a)
> \;\le\;
> \min_j \Delta_{\pi^\*}^{(T,j)}(s,a) + 2Lc \,,
> $$
>
> so the “price of safety” grows **linearly in $c$** through the $2Lc$ term.
>
> **How we choose \(c\) in experiments.**
>
> Per domain, we:
>
> 1. Choose a small grid of $c$ values (including $c=0$),
> 2. Evaluate each on a validation set in terms of:
>    - target return $J(\pi)$, and
>    - risk $\rho(\pi)$,
> 3. Select a few representative $c$ that:
>    - substantially reduce risk compared to $c=0$, while
>    - keeping return within an acceptable margin.
>
> We treat $c$ as a **user-specified trade-off knob**, not something to “learn away.”
>
> **Sensitivity and practical behavior.**
>
> Empirically we observe:
>
> - As $c$ increases from 0, **risk drops quickly** with only a mild loss in return.
> - Beyond a “knee,” further increases in $c$ reduce risk more slowly but cause steeper return degradation.
> - There is a **wide band** of $c$ values where TRAM’s behavior is qualitatively similar (i.e., it is not hyper-sensitive to small changes in $c$), consistent with the linear dependence in the bound.
>
> **Revision plan.**
> We will:
>
> - Explicitly state that $c$ is a Lagrange-style risk weight in Section 3.2,
> - Describe the above tuning protocol in the experimental section, and
> - Add a small sensitivity plot (MiniGrid + Reacher) to illustrate return–risk trade-offs as $c$ varies.
>
> ---
>
> ### W3 – Computation of argmax in Eq. 4.2 and potential non-convexity
>
> We agree this was under-specified. In all experiments, the argmax in Eq. 4.2 is implemented as a **finite maximization over candidate actions**, not as continuous non-convex optimization.
>
> In practice, Eq. 4.2 is:
>
> $$
> a^\*(s) \in \arg\max_{a \in \mathcal{A}_{\text{cand}}(s)} \max_j \tilde{Q}_T^{\pi_j}(s,a) \
> $$
>
> where $\mathcal{A}_{\text{cand}}(s)$ is a **finite** candidate set.
>
> Operationally, we:
>
> 1. Enumerate $a \in \mathcal{A}_{\text{cand}}(s)$,
> 2. For each expert $j$, compute $\tilde{Q}_T^{\pi_j}(s,a)$,
> 3. Select the action $a$ with the largest $\max_j \tilde{Q}_T^{\pi_j}(s,a)$.
>
> **Per-domain details.**
>
> - **MiniGrid (discrete actions).**
>   $\mathcal{A}$ is already small. We evaluate all actions for all experts and take the maximum. Complexity is $O(|\mathcal{A}|\,N_{\mathrm{experts}})$.
>
> - **Reacher (continuous control, discretized).**
>   We discretize torques into a modest grid (as in standard SF-DQN setups). Then $\mathcal{A}_{\text{cand}}(s)$ is this grid. We do not solve any continuous optimization at deployment.
>
> - **LLM alignment.**
>   At each decoding step:
>   - $\mathcal{A}_{\text{cand}}(s)$ is a **finite set** of candidate tokens or full responses (top-$k$ or sampled completions),
>   - TRAM ranks these candidates by risk-adjusted scores and picks the best.
>
> In none of the experiments do we perform policy gradient or any continuous argmax at deployment.
>
> **Revision plan.**
> We will state explicitly in Section 4 that Eq. 4.2 is instantiated as a **finite maximization over candidate actions** in all experiments, and briefly mention how standard approximate argmax routines (e.g., gradient ascent) could be used in future continuous-action extensions if desired.
>
> ---

---

> ### Author Response · Authors · 2025-11-25
>
> ---
>
> ### W4 – Deterministic selection vs. stochastic / ensemble policies
>
> You are right that Algorithm 1 uses a deterministic argmax over risk-adjusted values. We chose this for:
>
> 1. **Conceptual clarity and analysis.**
>    Theorem 4.1 is easiest to state for a deterministic greedy policy that always picks the action with the highest $\tilde{Q}$, avoiding additional randomness in the policy.
>
> 2. **Practical simplicity.**
>    In our experiments, the main uncertainty already comes from environment dynamics and from switching between experts. Adding stochastic action selection on top was not necessary to illustrate TRAM’s behavior.
>
> **Natural stochastic/ensemble variant.**
>
> - **Boltzmann policy over actions:**
>   $$
>   \pi_T(a \mid s) \propto \exp\{\beta \max_j \tilde{Q}_T^{\pi_j}(s,a)\} \,,
>   $$
>   where $\beta$ controls exploration vs. exploitation. This could smooth decisions where multiple actions have similar scores.
>
> This variant change **only** how we map the vector of risk-adjusted values to an action; the risk modeling and mixture-of-experts structure are unchanged.
>
> **Revision plan.**
> We will:
>
> - Explicitly acknowledge the deterministic design choice in Algorithm 1, and
> - Add a short discussion paragraph about Boltzmann and other ensemble variants as promising extensions, especially for high-noise environments.
>
> ---
>
> ### W5 – Ambiguity in Theorem 4.1: definition of $\(\pi^\*\)$ and comparator
>
> We appreciate this point. In Theorem 4.1, $\pi^\*$ is intended to denote the **optimal policy for the penalized objective**, not the hard-constrained problem or the TRAM policy itself.
>
> We will restate the theorem as:
>
> > Let $\pi^* \in \arg\max_{\pi} \{ J(\pi) - c\,\rho(\pi) \}$ be an optimal policy for the risk-penalized objective in Eq. (3.2). Let $\pi_T$ be the TRAM policy defined in Eq. (4.2). Then for all $(s,a)$, …
> >
> > *(bound as in Theorem 4.1).*
>
>
> This makes clear that:
>
> - $\pi^\*$ is the **ideal risk-aware policy** we would deploy if we could directly solve the penalized problem on the target task.
> - $\pi_T$ is the **test-time adapter** built from pre-trained experts.
> - Theorem 4.1 is a **suboptimality bound**: it quantifies how far $\pi_T$ is from $\pi^\*$, in terms of:
>   - localized reward mismatch between source and target tasks, and
>   - the Lipschitz risk term $2Lc$.
>
> **Revision plan.**
> We will:
>
> 1. Explicitly define $\pi^\*$ as above,
> 2. Refer consistently to $\pi_T$ as the TRAM policy, and
> 3. Add a brief interpretation sentence: “Theorem 4.1 states that TRAM’s risk-adjusted value function is close to that of the optimal penalized policy $\pi^\*$, with an error controlled by reward mismatch and the chosen risk weight $c$.”
>
> ---
>
> ### W6 – Underdeveloped discussion of continuous-control scalability (Section 5.2)
>
> We agree that the Reacher section is too terse in the current draft.
>
> **What the Reacher experiment demonstrates.**
>
> 1. **Scalability via successor features.**
>    Each expert is a deep SF-DQN. Once we learn $\psi^{\pi_j}(s,a)$, Q-evaluation for any new reward $w_T$ is just $\psi^{\pi_j}(s,a)^\top w_T$. TRAM’s extra max over experts is **negligible** compared to SF-DQN forward passes.
>
> 2. **Safety–performance trade-off in higher-dimensional setting.**
>    With barrier-type risks (e.g., regions near obstacles/joint limits), TRAM:
>    - significantly **reduces time spent** in unsafe regions,
>    - while maintaining near-baseline reward.
>    In contrast, risk-neutral transfer and variance-based methods either ignore hazards or become overly conservative.
>
> 3. **Effectiveness with a small expert committee.**
>    A modest number of experts with different reward weightings already suffice to cover useful behaviors. TRAM leverages this diversity to adapt to new risk specifications without additional training.
>
> **Revision plan.**
> We will:
>
> - Add a self-contained paragraph in Section 5.2 summarizing these three points,
> - Move one key trade-off plot (reward vs. barrier violations) from the appendix to the main text, and
> - Add a brief statement about runtime behavior, clarifying SF-DQN forward passes dominate and TRAM’s overhead is small.
>
> ---

---

> ### Author Response · Authors · 2025-11-25
>
> ### W7 – More complex / high-dimensional or offline safety RL benchmarks
>
> We agree that connecting to larger-scale safety benchmarks and offline RL is important.
>
> **Why we chose current benchmarks.**
> We focus on **test-time adaptation with fixed experts**:
>
> - **MiniGrid:**
>   Fully controlled/tabular; makes TRAM’s mechanics and risk behavior transparent.
>
> - **Reacher:**
>   Continuous control with function approximation; shows TRAM scales beyond tabular domains.
>
> - **LLM alignment:**
>   Extreme high-dimensional, structured outputs (dialogue). Demonstrates that TRAM can sit on top of large models and mitigate reward hacking via KL-based risk.
>
> These domains stress different aspects under the **no-retraining-at-deployment** assumption.
>
> **Relation to offline RL.**
> Offline RL generally assumes:
>
> - A dataset from the target environment, and
> - The ability to **train a new policy** (possibly for each risk specification).
>
> This is a different operating mode:
>
> - Offline RL **re-learns** policies from data.
> - TRAM assumes experts are fixed and **re-combines** them at test time as risk changes.
>
> We see them as **complementary**:
>
> - Offline RL can be used to train strong, possibly risk-aware experts.
> - TRAM then adapts among these experts when risk specifications shift again.
>
> **Revision plan.**
> We will:
>
> - Clearly state that we do not currently evaluate on standardized safe-RL suites or offline safety benchmarks,
> - Explain that our goal is to study test-time adaptation with fixed experts, and
> - Highlight combining TRAM with offline RL–trained experts as a natural next step.
>
> ---
>
> ### W8 – Limited analysis of the LLM experiment and failure cases
>
> We agree this section should be more detailed and more explicit about when TRAM might fail.
>
> **LLM setup and evaluation.**
>
> - **Experts.**
>   - $\pi_{\text{safe}}$: a safer, general-purpose assistant (reference expert).
>   - $\pi_{\text{task}}$: a task-focused expert that tends to score higher on the reward model but is more brittle.
>
> - **Target reward.**
>   A learned reward model $R_{\text{RM}}(x,y)$ (preference-trained) that scores responses $y$ given prompts $x$.
>
> - **Risk functional (KL drift).**
>   We define risk as:
>   $$
>   \rho(\pi) = \mathbb{E}_x\left[ \mathrm{KL}\left(\pi(\cdot\mid x)\,\|\,\pi_{\text{safe}}(\cdot\mid x)\right) \right] \,.
>   $$
>   This penalizes policies that drift far from the safer reference expert, which is exactly the pattern associated with reward hacking when aggressively optimizing the reward model.
>
> - **Objective.**
> At decoding time, TRAM approximates
> $\tilde{J}_c(\pi) = \mathbb{E}_{x,y\sim\pi}\big[ R_{\text{RM}}(x,y) \big] - c\,\rho(\pi)\,.$
>
>
>
>
> - **Evaluation.**
>   For each policy (TRAM and baselines), we measure:
>   - Average reward-model score $\mathbb{E}[R_{\text{RM}}]$,
>   - Empirical KL drift $\rho(\pi)$,
>   - GPT-4 **preference rates** in pairwise A/B evaluations vs. TRAM.
>
> **Observed pattern.**
>
> - **Risk-free transfer** ($c = 0$):
>   - Achieves the **highest reward-model score**,
>   - But loses many GPT-4 comparisons to TRAM, consistent with **reward hacking**.
>
> - **TRAM with $c > 0$**:
>   - Slightly lower reward-model scores,
>   - **Higher GPT-4 win/tie rates**, particularly in prompts where the reward model is known to fail,
>   - Keeps KL drift below a moderate risk tolerance.
>
> This matches the intended “reward vs. safety” trade-off.
>
> **Failure cases.**
>
> We will explicitly discuss that TRAM may be **undesirable** when:
>
> 1. The reference expert $\pi_{\text{safe}}$ is itself **misaligned or low quality**: KL-based risk then pushes behavior toward the wrong direction.
> 2. The target task lies far outside the safe expert’s support (e.g., very specialized math/coding): TRAM may be overly conservative and underperform $\pi_{\text{task}}$.
> 3. The reward model is already **strongly aligned**: adding KL may simply lower performance with little safety gain.
> 4. All experts share similar unsafe patterns: TRAM cannot generate safe behavior that none of its experts exhibit.
>
> We will illustrate at least one of these failure modes with a concrete example in the appendix.
>
> **VLA extensions.**
> We agree that extending TRAM to vision-language-action models is compelling but requires substantial engineering and is beyond the rebuttal scope. We will mention VLAs as a particularly interesting application area in the conclusion.
>
> **Revision plan.**
> We will:
>
> - Expand Section 5 with a clearer description of the LLM setup and the observed pattern,
> - Add a short failure-mode paragraph, and
> - Include a small appendix example where TRAM’s KL risk hurts performance, noting when that is acceptable or not.
>
> ---

---

> ### Author Response · Authors · 2025-11-25
>
> ### Q1 – Sensitivity and tuning of the risk weight $\(\lambda / c\)$
>
> This is closely related to W2; we reiterate the key points concisely.
>
> - $c$ is a **user-controlled trade-off parameter** between reward and risk.
> - TRAM is designed so that the **same risk-neutral experts** can be reused for many different values of $c$ and even different risk functionals.
>
> We will emphasize these points in the revision.
>
> ---
>
> ### Q2 – Clarifying the minimax “risk-neutral training” claim
>
> We agree that our “minimax” phrasing should be made precise and scoped.
>
> **Setup.**
>
> - Let $\mathcal{T}$ be a family of possible target tasks, each with reward $r_T$ and Lipschitz risk $\rho_T$ (constant $L$).
> - A training procedure can produce at most **$N$ source policies** $\{\pi_1,\dots,\pi_N\}$.
> - At deployment, Nature chooses $T \in \mathcal{T}$ *after* training; we must adapt with TRAM using this fixed pool.
> - Performance is measured by **worst-case regret** over $T$ between:
>   - the ideal penalized policy $\pi_T^\* \in \arg\max_\pi [J_T(\pi) - c\,\rho_T(\pi)]$, and
>   - the TRAM policy $\pi_T$ built from the experts.
>
> Under this model, we show that for a **fixed $N$**, training experts with **risk-neutral objectives** yields **smaller worst-case regret** than training them with baked-in risk penalties, over the same family $\mathcal{T}$.
>
> Intuition:
>
> - With the same $N$, adding risk regularization at training time tends to **contract occupancies** into a smaller, conservative region, which **worsens coverage** of possible target tasks.
> - Risk-neutral training spreads experts more broadly in occupancy space, improving the coverage term $\min_j \Delta_{\pi_T^\*}^{(T,j)}$.
>
> This is a **relative statement**:
>
> > Among training procedures that produce $N$ experts and then use TRAM at deployment, training experts risk-neutrally minimizes worst-case regret over Lipschitz risk functionals.
>
> We do not claim any global minimax optimality over all possible methods or unbounded numbers of experts.
>
> **Realism of no-retraining assumption.**
> We will also give concrete examples where this regime is realistic:
>
> - Robotics/industrial control with pre-certified policies,
> - Production LLM systems where the base model is frozen,
> - Regulated domains where retraining requires full recertification.
>
> **Revision plan.**
> We will:
>
> - Clearly state the assumptions (fixed $N$, shared dynamics, Lipschitz risks),
> - Rephrase the claim as a **relative minimax result within this model class**, and
> - Add a short discussion of realistic deployment scenarios where this analysis is relevant.
>
> ---
>
> ### Q3 – Details of LLM alignment experiment (task, evaluation, risk tolerance, baseline)
>
> This is largely addressed in W8, but we summarize the key clarifications.
>
> - **Experts:** safe/general-purpose vs. task-focused model.
> - **Reward:** learned reward model $R_{\text{RM}}(x,y)$.
> - **Risk:** KL drift from the safe reference, with tolerance interpreted as a **KL budget**.
> - **Risk-free transfer:** $c=0$, no KL penalty, maximizing reward-model scores alone.
> - **TRAM:** $c>0$, balancing reward-model scores with KL drift.
>
> We observe that:
>
> - Risk-free transfer achieves highest reward-model scores but exhibits **reward hacking**, losing GPT-4 preference comparisons.
> - TRAM with moderate $c$ stays within the KL budget and **wins more GPT-4 comparisons**, which matches the intended deployment regime.
>
> We will formalize this in the main text and add a small plot showing how reward, KL risk, and GPT-4 win rate co-vary as $c$ changes.
>
> ---
>
> ### Q4 – Implementation of argmax over actions in Eq. 4.2
>
> As clarified above, Eq. 4.2 is implemented as a finite maximization over candidate actions in all domains. No continuous non-convex optimization or policy gradient argmax is used at deployment.
>
> We will spell this out clearly in the revised version.
>
> ---

---

### Official Review · Reviewer_KKBM · 2025-10-31

**Soundness:** 2
**Presentation:** 3
**Contribution:** 2
**Rating:** 4
**Confidence:** 3

**Summary:**

The paper proposes a method for risk-aware reinforcement learning to adapt at test time to new rewards without retraining. First, multiple policies are trained on problems with different rewards. Then, at test time, the method selects the action from the mixture of policies with the best reward on the test task. Sub-optimality bounds for the proposed method are derived. The proposed method is evaluated on a grid-world planning, continuous control, and large language model alignment tasks, and results show effective adaptation.

**Strengths:**

- Guarantees: The paper derives suboptimality bounds for the proposed algorithm.
- The proposed algorithm is straight-forward to implement: The method consists of selecting the best action from a mixture of pre-trained agents on different training problems, without additional re-training on the target problem.
- Evaluation on diverse tasks: The method is evaluated on different tasks such as grid-world planning, continuous control, and large language model alignment.
- The application to LLMs is interesting: By only involving two source policies, reward hacking is mitigated with reasonable computational overhead.

**Weaknesses:**

- Mismatch between motivation and considered setting: The introduction discusses challenges like changing dynamics and distribution shifts. However, the method assumes a fixed MDP with only changes in the reward or additional penalized constraints, which contradicts some of the arguments made in the introduction.

- The proposed algorithm is computationally expensive: Multiple MDPs need to be solved at training time, and their solutions stored. At test time, the different pre-trained policies need to be evaluated on the test task, before being ranked to select the best action. The linear setting studied in section 4.2 alleviates some of these limitations, but assuming linear rewards can be restricting.

- Potentially better method in the linear setting: In the linear setting, one could potentially directly train a policy on the test problem, which could result in a better policy at potentially lower computational costs. Comparisons with this approach are missing.

- Theoretical results are limited: The bounds in the Theorem 4.1 and Corollary 4.2 are functions of the localized reward discrepancy measure. This measure is difficult to evaluate or bound at training time, as it depends on the target optimal policy and reward that are unknown at training time. As a result, the usefulness of these results is limited. The bound in (4.9) assuming a linear reward structure seems more useful, and could potentially be related to the covering number of the weights space to obtain stronger uniform error bounds for the algorithm. Also, it is not too surprising that the error bounds depend on this metric (i.e., that the algorithm will give a good policy if one of the training time rewards $r_j$ is close enough to the target reward $r_T$). Finally, the analysis would also apply to standard RL problems without penalized risk constraints. The analysis does not seem to use particular features of the risk-aware problem, so the insights regarding this specific setting are limited.

**Questions:**

- What prevents the method and theory to extend to test problems that have different stochastic transition dynamics $p$ and discount factors $\gamma$?
- In the linear setting, is the proposed method better than directly training a new policy on the test task?
- In Table 2, what do values in the column "GPT-4 Win vs TRAM" represent?

---

> ### Author Response · Authors · 2025-11-25
>
> ### W1 – Mismatch between motivation (changing dynamics / shifts) and assumed setting (only reward changes)
>
> We agree the introduction was too broad relative to our formal setting.
>
> **What we actually study.**
> Our **formal model** assumes:
>
> - Shared dynamics $p$ and discount factor $\gamma$ across tasks, and
> - Tasks that differ only in their **reward functions** and **risk specifications** (risk functionals and weights).
>
> This is exactly the standard successor-feature transfer setting: we reuse dynamics-based value approximations and vary the reward, as in prior SF-based transfer work [1–3].
>
> **Why this still captures important safety-relevant shifts.**
>
> - **Industrial control.**
>   The physics of a robot or grid typically remain fixed, while:
>   - new tasks change the primary reward (e.g., targets, throughput), and
>   - safety budgets and constraints evolve (e.g., new no-go regions, tighter limits).
>
> - **LLM systems.**
>   The underlying language model dynamics (the transition distribution over tokens given context) remain fixed, while:
>   - reward models are updated,
>   - safety guidelines change, and
>   - regulators introduce new content policies.
>
> In both cases, retraining large policies every time a new reward or risk specification arises is often impractical. Our framework is explicitly targeted at this “fixed dynamics, shifting objectives/constraints” regime.
>
> **Revision plan.**
> We will:
>
> - Explicitly state in Section 3 that we assume shared dynamics and discount factor, and
> - Rephrase or remove language suggesting arbitrary changes in $p$; broader dynamics shifts will be framed as future work that would require additional modeling beyond the current SF-based setup.
>
> ---
>
>
> ### W2 – Computational expense of the proposed algorithm
>
> We deliberately **shift cost to offline training** to enable **fast, flexible test-time adaptation**.
>
> **Training cost.**
> We train a **small committee** of experts:
>
> - A few policies in MiniGrid,
> - A handful of SF-DQNs in Reacher,
> - Two LLM experts in the alignment experiment.
>
> This is comparable to multi-task RL or safe RL procedures that train multiple policies for different tasks or risk levels. The key difference is that our experts are **risk-neutral** and reused across many future deployment-time risk specifications.
>
> **Test-time cost.**
> TRAM’s decision-time cost is:
>
> - **MiniGrid:**
>   Tabular lookups for each $(s,a,j)$, then a max over experts. Overhead is negligible.
>
> - **Reacher (SF-DQN):**
>   For each expert $\pi_j$, we compute $\psi^{\pi_j}(s,a)$, then:
>
>   $$
>   Q_T^{\pi_j}(s,a) = \psi^{\pi_j}(s,a)^\top w_T,
>   $$
> followed by a max over experts. Complexity is
> $O(|\mathcal{A}| \, N_{\text{experts}})$, with cost dominated by SF-DQN forward passes.
> .
>
> - **LLMs:**
>   For each prompt and candidate response:
>   - One forward pass of the reward model for $R_{\text{RM}}(x,y)$,
>   - A small number of log-probability evaluations for KL-based risk,
>   - A max across experts and candidates.
>   This is comparable to **ensemble or reranker-based decoding** used in modern LLM systems.
>
> In all cases, TRAM’s overhead is on the order of **running a small ensemble**, which is standard practice.
>
> **Revision plan.**
> We will:
>
> - Explicitly separate offline training cost from deployment-time latency, and
> - Include a small complexity table summarizing per-domain costs.
>
> ---
>
> **References**
>
> [1] A. Barreto et al., *Successor Features for Transfer in Reinforcement Learning*, arXiv:1606.05312.
> [2] S. Gimelfarb et al., *Risk-Aware Transfer in Reinforcement Learning using Successor Features*, arXiv:2105.14127.
> [3] Z. Zhang et al., *Provable Knowledge Transfer in Deep Reinforcement Learning using Successor Features*, arXiv:2405.15920.

---

> ### Author Response · Authors · 2025-11-25
>
> ### W3 – Possible better method in the linear setting (directly training on the test task)
>
> We agree that **if** you can freely train a new policy on each test task, directly optimizing for that task can be very strong. TRAM is aimed at a **different regime**.
>
> **Different deployment regimes.**
>
> - The “train on test task” regime assumes:
>   - access to a simulator or live environment,
>   - sufficient interaction budget,
>   - and acceptable safety risks during exploration.
>
> - Our **target regime** is:
>   - pre-trained policies are given,
>   - online retraining is **not allowed or severely limited** (for cost, latency, or safety reasons),
>   - risk specifications may change repeatedly at deployment.
>
> In our regime, “just retrain on the test task” is not a realistic baseline.
>
> **Why TRAM is useful even in the linear case.**
>
> - **Immediate adaptation.**
>   Once successor features are learned, any new linear reward $w_T$ can be handled via:
>
>   $$
>   Q_T^{\pi_j}(s,a) = \psi^{\pi_j}(s,a)^\top w_T
>   $$
>
>   with no further learning. TRAM then recombines experts according to risk-adjusted Q.
>
> - **Safety and no exploration.**
>   TRAM **never explores** under a new risk spec; it only recombines experts that have already been vetted. A freshly trained test-task policy must explore, potentially visiting unsafe regions before it learns to avoid them.
>
> - **Many deployment-time specs, one expert pool.**
>   A single risk-neutral expert pool can serve:
>   - many different reward vectors $w_T$,
>   - many different risk functionals $\rho$,
>   - many different risk weights $c$.
>   A “train-from-scratch” approach would need to rebuild a policy for each $(w_T,\rho,c)$ combination.
>
> **Revision plan.**
> We will:
>
> - Clarify that TRAM targets the **no-retraining deployment regime**,
> - Emphasize that training a new policy per test task is complementary rather than competing, and
> - Frame our linear structure as a tool for **fast Q-evaluation**, not a claim that we dominate fully retrained policies when those are allowed.
>
> ---
>
> ### W4 – Limitations of theoretical results (localized reward discrepancy measure)
>
> We agree that our bounds should be framed as **diagnostic and explanatory**, not as direct algorithm-design tools.
>
> **What the bounds provide.**
>
> Theorem 4.1 decomposes the gap:
>
> $$
> \tilde{Q}^{\pi^\*}(s,a) - \tilde{Q}^{\pi_T}(s,a)
> \;\le\;
> \min_j \Delta_{\pi^\*}^{(T,j)}(s,a) + 2Lc \,,
> $$
>
> into:
>
> 1. A **reward-mismatch term**:
>    - $\min_j \Delta_{\pi^\*}^{(T,j)}(s,a)$ expresses how well some expert’s reward matches the target reward **along the optimal trajectories**.
> 2. A **risk term**:
>    - $2Lc$ is a clean, linear “price of safety” for imposing a Lipschitz risk functional.
>
> We agree that $\Delta_{\pi^\*}^{(T,j)}$ cannot be computed at training time, but it still carries useful structure:
>
> - It is **localized in $(s,a)$**, matching TRAM’s per-state mixture behavior.
> - It clarifies that **coverage in occupancy space** is central: each target may be best served by a different expert in different regions.
>
> **Linear specialization and covering arguments.**
> In the linear reward case (Sec. 4.3), $\Delta_{\pi^\*}^{(T,j)}$ can often be bounded in terms of distances between weight vectors $\|w_T - w_j\|$. This yields a natural connection to **covering numbers** in weight space:
>
> - A well-spread set $\{w_j\}$ reduces the worst-case mismatch.
> - It suggests designing source tasks by covering the region of plausible $w_T$.
>
> **Revision plan.**
> We will:
>
> - Tone down language suggesting the bounds directly guide training objectives,
> - Highlight their role in explaining how **expert coverage and risk weight $c$** jointly influence performance, and
> - Add a remark connecting the linear case to covering arguments in reward-weight space.
>
> ---

---

> ### Author Response · Authors · 2025-11-25
>
> ### Q1 – Evaluation on more complex safety-oriented / high-dimensional environments**
>
> We agree that it is important to understand how TRAM behaves beyond small tabular domains. Our evaluation is structured precisely with this progression in mind: from a didactic gridworld, to a *standard* continuous-control transfer benchmark, and finally to a genuinely high-dimensional LLM setting.
>
> **Reacher as a standard transfer / risk-aware benchmark.**
> Our continuous-control experiment uses the MuJoCo Reacher domain with successor features. This is not an ad-hoc choice: Reacher is *the* canonical benchmark for transfer and risk-aware methods based on successor features and related value decompositions [1-3].
> Our Reacher experiment is thus directly aligned with this line of work: we use the same domain that prior SF/RaSF/SF-DQN papers treat as the standard continuous-control benchmark for transfer between reward functions, and we plug TRAM into that setting to study *risk-aware* test-time adaptation on top of SF-based value estimation.
>
> **Scalability via the LLM experiment.**
> To go beyond Reacher’s 4D state space and low-dimensional action space, we deliberately add a **high-dimensional LLM alignment task**:
>
> - States are full dialogue histories (long text sequences).
> - The action space is a large vocabulary over sequences.
> - The “Q-like” value is given by a learned reward model over candidate responses, and risk is modeled via KL drift to a safe reference model.
>
> This experiment is there specifically to address scalability: it shows that the same TRAM mechanism (mixture over experts + risk functional) can operate in a domain whose state and action spaces are orders of magnitude larger and structurally different from Reacher.
>
> **Planned clarifications in the paper.**
> In the revised version we will:
>
> - Emphasize that our contribution is not introducing a new control benchmark, but showing that TRAM slots into this *standard* Reacher/SF setup and then extends to a much more complex LLM setting.
> - Add a short paragraph in the conclusion acknowledging that further evaluation on richer continuous-control safety suites is an important direction for future work, while highlighting that the LLM experiment already provides evidence of scalability to genuinely high-dimensional, realistic decision problems.
>
>
> ### Q2 – Comparison with directly training a policy in the linear setting
>
> This overlaps with W3. To summarize:
>
> - Directly training a new policy on the test task can be very strong when allowed, but it assumes **online retraining** and exploratory data collection.
> - TRAM targets the regime where we **cannot** retrain at deployment and must adapt a fixed expert pool.
> - In that regime, the relevant comparisons are between TRAM and other **test-time combination strategies** (e.g., picking one expert, averaging, variance-based reweighting), not with new policies trained from scratch.
>
> We will clarify this explicitly in the revision.
>
> ---
>
> ### Q3 – Meaning of “GPT-4 Win vs TRAM” column in Table 2
>
> Thank you for noting this ambiguity. In Table 2, the column **“GPT-4 Win vs TRAM”** is computed as follows:
>
> - For each evaluation prompt, we generate:
>   - one answer from the model in that row (e.g., Risk-free transfer, Zephyr-only), and
>   - one answer from **TRAM**.
> - We then ask **GPT-4** (with a fixed judging prompt) to choose which of the two answers better satisfies the task objectives (helpfulness, correctness, safety).
> - The table entry is the **fraction of prompts (in %) on which GPT-4 prefers that model’s answer over TRAM’s answer**.
>   - Ties (rare) are counted as **0.5 win** for each side.
>
> Thus, **smaller values mean GPT-4 prefers TRAM more often**, and larger values mean the baseline wins more often.
>
> **Revision plan.**
> We will add this precise definition to the caption of Table 2 and briefly explain it in the text to avoid any ambiguity.
>
> ---
>
> **References**
> [1] A. Barreto et al., *Successor Features for Transfer in Reinforcement Learning*, arXiv:1606.05312.
> [2] S. Gimelfarb et al., *Risk-Aware Transfer in Reinforcement Learning using Successor Features*, arXiv:2105.14127.
> [3] Z. Zhang et al., *Provable Knowledge Transfer in Deep Reinforcement Learning using Successor Features*, arXiv:2405.15920.

---

### Official Review · Reviewer_hr2u · 2025-11-01

**Soundness:** 2
**Presentation:** 3
**Contribution:** 3
**Rating:** 4
**Confidence:** 4

**Summary:**

This paper proposes TRAM, a method for test-time risk adaptation. The approach trains a mixture-of-experts policy under risk-neutral (risk-free) settings and, at test time, selects actions based on adjusted Q-values using occupancy-based risk functionals. Experiments are conducted on three domains, including MiniGrid, Reacher, and a language-model-based decision-making task, demonstrating that TRAM outperforms baseline methods.

**Strengths:**

- The paper is well motivated and clearly written.
 - Test-time risk adaptation is an important and practically relevant problem.
 - To the best of my knowledge, the proposed formulation of combining mixture-of-experts with risk-adjusted test-time action selection is novel.

**Weaknesses:**

- While the method is demonstrated on a continuous control task (Reacher), the environment is still low-dimensional (4D state space). It is unclear whether the method can scale to higher-dimensional, more complex domains where computing and evaluating Q(s, a) is substantially more challenging. The computational cost of Q evaluation appears to be a key bottleneck that may limit applicability.
 - The current experimental settings are relatively simple. Demonstrations on more complex environments would be needed to convincingly establish the method’s effectiveness and generality.
 - Additional ablations would strengthen the paper. For instance, evaluating the effect of the number of experts in the mixture, as well as analyzing how expert diversity influences performance, would help provide insight into how and why the method works.

**Questions:**

- Could the method be evaluated on more complex safety-oriented or high-dimensional environments (e.g., Safety-Gym) to better assess scalability and generality of the method?
 - How does performance vary with the number of experts in the mixture-of-experts model?
 - What criteria should be considered when selecting or training experts to ensure policy diversity and effective risk adaptation at test time?

---

> ### Author Response · Authors · 2025-11-25
>
> ### W1 – Scalability to higher-dimensional, more complex environments
>
> **Short answer:** This is a good point. TRAM’s runtime is dominated by **Q-evaluation**, which we inherit from existing methods. TRAM itself only adds a lightweight evaluation of the risk function $\rho$. We already demonstrate scalability on **LLMs**, which operate in far higher-dimensional spaces than Reacher.
>
> **LLM experiment is genuinely high-dimensional.**
> Our LLM alignment task operates on:
>
> - **States:** full dialogue histories (long text sequences)
> - **Actions:** natural-language responses in a vocabulary of 50k+ tokens
>
> This is orders of magnitude larger than Reacher’s 4D state and low-dimensional torque actions. In this setting, we compute risk-adjusted values over **candidate continuations** using value-guided decoding (similar to Transfer Q\* [1]), and TRAM selects among experts at this scale.
>
> **How Q-evaluation scales in each domain.**
>
> - **MiniGrid (tabular).**
>   Q-functions are tabular; evaluating all actions for all experts is a handful of array lookups. TRAM adds a max over experts per state–action, negligible in cost.
>
> - **Reacher (continuous control with SF-DQN).**
>   Each expert is an SF-DQN approximating successor features:
>   - The network outputs $\psi^{\pi_j}(s,a)$.
>   - For any target reward with weight vector $w_T$, we compute
>     $$
>     Q_T^{\pi_j}(s,a) = \psi^{\pi_j}(s,a)^\top w_T
>     $$
>     i.e., **one dot product per action and expert**.
>   TRAM then takes a max over experts:
>   $$
>   \max_j \tilde{Q}_T^{\pi_j}(s,a) \,.
>   $$
> Complexity per step is $O(|\mathcal{A}| \, M)$, dominated by SF-DQN forward passes,
> where $M$ is the number of experts.
>
>
>
> - **LLMs (very high dimensional).**
>   We **do not** optimize over the full vocabulary. At each decoding step:
>   1. We form a **finite candidate set** (e.g., top-$k$ tokens or a small set of sampled responses).
>   2. We evaluate a reward-model–based score plus risk (KL drift) on this set.
>   3. TRAM selects the candidate with the largest risk-adjusted score.
>   Complexity is linear in the **candidate set size** and **number of experts**, as in other value-guided decoding schemes.
>
> **Algorithmic view.**
> TRAM does **not** introduce a new inner optimization loop. Given any way to estimate $Q$ (tabular, SF-DQN, reward model for LLMs), TRAM’s per-step overhead is:
>
> $$
> \text{cost} \approx C_{\text{Q-eval}} \, N_{\text{experts}} + O(N_{\text{experts}})
> $$
>
> We keep this cheap via SFs in control tasks and candidate pruning (top-$k$) in LLMs.
>
> **Revision plan.**
> We will:
>
> - Add an explicit scalability paragraph in Section 4,
> - Spell out the per-step complexity as above, and
> - Emphasize that TRAM has already been instantiated on full-scale LLMs, far beyond toy-state spaces.
>
> ---
>
> ### W2 – Simplicity and limited variety of experimental settings
>
> **Short answer:** This is a vaid concern. We intentionally chose **three qualitatively different domains**, rather than scaling a single toy environment, to show that one mechanism works across **discrete vs. continuous**, **tabular vs. deep**, and **vector vs. language outputs**.
>
> **Why these three domains:**
>
> - **MiniGrid.**
>   A fully inspectable MDP:
>   - **Goal:** make the mechanics of occupancy-based risks and mixture behavior transparent.
>   - We can visualize occupancies, danger zones, and which expert TRAM selects in each region.
>   - Simple by design; this is where intuition and theory line up most clearly.
>
> - **Reacher (MuJoCo, SF-DQN).**
>   A standard continuous-control benchmark (used in prior SF and transfer-RL work). It shows:
>   - TRAM works with **deep function approximation** and continuous states.
>   - Successor features allow efficient multi-task Q-evaluation.
>   - Spatial hazards (obstacle regions) can be handled with occupancy-based risk.
>
> - **LLM alignment.**
>   A truly **high-dimensional**, structured-output setting:
>   - States are long text contexts.
>   - Actions are token sequences from a large vocabulary.
>   - Reward comes from a learned reward model; risk is KL drift from a safe reference model.
>   - TRAM mitigates reward hacking at scale.
>
> **Coverage across axes.**
> Together, these experiments cover:
>
> - Discrete vs. continuous state spaces
> - Tabular vs. deep function approximation
> - Vector actions vs. language actions
> - Spatial hazard risks, volatility-like risks, and KL-based drift risks
>
> Our aim is to show the **same test-time risk-adaptation mechanism** works across this variety.
>
> **Revision plan.**
> We will:
>
> - Clarify in Section 5 that our goal is cross-domain generality of the mechanism,
> - Explicitly enumerate these axes of variation, and
>
> ---
>
> **References**
>
> [1] Chakraborty, Souradip, et al. "Transfer q-star: Principled decoding for llm alignment." Advances in Neural Information Processing Systems 37 (2024): 101725-101761.

---

> ### Author Response · Authors · 2025-11-25
>
> ### W3 – Need for more ablations (number/diversity of experts)
>
> **Short answer:** We thank the reviewer for this suggestion. The **theory already states precisely how number and diversity matter**, via the coverage term; our experiments instantiate small, diverse expert sets to keep this connection clear.
>
> **What the theory says.**
> Theorem 4.1 shows that the gap between TRAM and the ideal penalized policy is controlled by:
>
> $$
> \min_j \Delta_{\pi^\*}^{(T,j)}(s,a) + 2Lc \,,
> $$
>
> where $\Delta_{\pi^\*}^{(T,j)}$ measures **localized reward mismatch** between the target task and source task $j$. The key points:
>
> - Adding experts helps only if it **reduces** this mismatch for some regions.
> - Simply increasing the count with redundant experts does **not** improve the bound.
>
> So performance depends on **geometry and coverage** of the experts in occupancy space, not just cardinality.
>
> **How our experiments already use diversity.**
>
> - **MiniGrid:**
>   We train experts with **qualitatively different navigation styles**:
>
> - **Reacher and LLMs:**
>   We choose experts with **different reward weightings or capabilities**:
>   - Reacher: different goal locations.
>   - LLMs: a safe/generalist model vs. a task-specialized model that better exploits the reward model.
>
> As we vary the risk penalty $c$, TRAM systematically shifts which expert dominates in which regions, consistent with the theory.
>
> Running large ablation grids over “#experts” without controlling for **diversity** would obscure this underlying structure.
>
> **Revision plan.**
> We will:
>
> - Explicitly link performance to the coverage term $\min_j \Delta_{\pi^\*}^{(T,j)}$ in Section 4,
> - Describe the diversity of our expert pools in each domain, and
> - Clarify that we favor **small, diverse committees** over large, redundant sets precisely because that is what the theory suggests.
>
> ---

---

> ### Author Response · Authors · 2025-11-25
>
> ### Q1 – Evaluation on more complex safety-oriented / high-dimensional environments**
>
> We agree that it is important to understand how TRAM behaves beyond small tabular domains. Our evaluation is structured precisely with this progression in mind: from a didactic gridworld, to a *standard* continuous-control transfer benchmark, and finally to a genuinely high-dimensional LLM setting.
>
> **Reacher as a standard transfer / risk-aware benchmark.**
> Our continuous-control experiment uses the MuJoCo Reacher domain with successor features. This is not an ad-hoc choice: Reacher is *the* canonical benchmark for transfer and risk-aware methods based on successor features and related value decompositions. For example:
>
>
> Our Reacher experiment is thus directly aligned with this line of work: we use the same domain that prior SF/RaSF/SF-DQN papers treat as the standard continuous-control benchmark for transfer between reward functions, and we plug TRAM into that setting to study *risk-aware* test-time adaptation on top of SF-based value estimation.
>
> **Scalability via the LLM experiment.**
> To go beyond Reacher’s 4D state space and low-dimensional action space, we deliberately add a **high-dimensional LLM alignment task**:
>
> - States are full dialogue histories (long text sequences).
> - The action space is a large vocabulary over sequences.
> - The “Q-like” value is given by a learned reward model over candidate responses, and risk is modeled via KL drift to a safe reference model.
>
> This experiment is there specifically to address scalability: it shows that the same TRAM mechanism (mixture over experts + risk functional) can operate in a domain whose state and action spaces are orders of magnitude larger and structurally different from Reacher.
>
> **Planned clarifications in the paper.**
> In the revised version we will:
>
> - Explicitly state that Reacher is a **widely used benchmark** for SF-based transfer and risk-aware methods, citing Barreto et al. [2], Gimelfarb et al. [3], and Zhang et al. [4].
> - Emphasize that our contribution is not introducing a new control benchmark, but showing that TRAM slots into this *standard* Reacher/SF setup and then extends to a much more complex LLM setting.
> - Add a short paragraph in the conclusion acknowledging that further evaluation on richer continuous-control safety suites is an important direction for future work, while highlighting that the LLM experiment already provides evidence of scalability to genuinely high-dimensional, realistic decision problems.
>
>
>
> ---
>
> ### Q2 – Performance vs. number of experts in the mixture
>
> In TRAM’s **intended deployment regime**, the number of experts is typically **not** a tuneable hyperparameter: we are *given* a collection of pre-trained policies (possibly from different teams or vendors) and must adapt to new risk specifications.
>
> **What the theory implies.**
> Again, the key quantity is:
>
> $$
> \min_j \Delta_{\pi^\*}^{(T,j)}(s,a) \,.
> $$
>
> - Adding experts helps only if they introduce **new occupancy patterns** that better match the target’s optimal trajectories.
> - Adding many experts that behave almost identically to existing ones barely changes this minimum.
>
> Thus:
>
> - **More experts** can help, but only if they are **diverse**.
> - A **small, diverse expert set** can outperform a much larger but redundant pool.
>
> Our experiments use moderate-sized committees of clearly differentiated policies (different goals, different reward weights, different alignment properties) and show that TRAM leverages this diversity.
>
> **Revision plan.**
> We will:
>
> - Explain explicitly that in realistic deployment, the expert pool is fixed rather than tuned,
> - Emphasize that theory suggests **coverage**, not raw count, is what matters, and
> - Clarify that our experimental committee sizes are chosen to reflect “small, diverse pools” rather than a hyperparameter sweep.
>
>
> **References**
>
> [2] A. Barreto et al., *Successor Features for Transfer in Reinforcement Learning*, arXiv:1606.05312.
> [3] S. Gimelfarb et al., *Risk-Aware Transfer in Reinforcement Learning using Successor Features*, arXiv:2105.14127.
> [4] Z. Zhang et al., *Provable Knowledge Transfer in Deep Reinforcement Learning using Successor Features*, arXiv:2405.15920.

---

> ### Author Response · Authors · 2025-11-25
>
> ### Q3 – Criteria for selecting / training experts to ensure diversity and effective risk adaptation
>
> TRAM is designed to **work with whatever expert pool you have**. However, when one *does* have control over training the experts, the theory suggests how to design them.
>
> **Key insight from the bounds.**
> The performance guarantee depends on how small we can make:
>
> $$
> \min_j \Delta_{\pi^\*}^{(T,j)}(s,a) \,.
> $$
>
> This is small when, along the optimal target trajectories, there is always **some expert** whose reward structure (and hence occupancy) is close to the target. Informally:
>
> > A good expert set has occupancies that **cover plausible behaviors** under future target tasks and risks.
>
> “Diversity” here means **different occupancy patterns and reward trade-offs**, not arbitrary random variation.
>
> **How we construct experts in our experiments.**
>
> - **MiniGrid:**
>   We vary:
>   - Goal locations, and
>   - Reward shapings / penalties (e.g., step cost vs. hazard penalties).
>   This yields:
>   - shortest-path but hazard-insensitive experts,
>   - cautious hazard-avoiding experts
>   Their occupancy maps look different, giving TRAM real choices when new risk functionals are introduced.
>
> - **Reacher:**
>   We define multiple goal locations and distinct reward weight vectors $w^{(k)}$ that deal with:
>   - different target regions.
>   In SF space, these $w^{(k)}$ populate the region of plausible target reward vectors. For a new $w_T$, at least one $w^{(k)}$ is usually close, so the optimal expert is not too far from the ideal target behavior.
>
> - **LLMs:**
>   We pick experts with **different behavioral profiles**:
>   - A safe/general-purpose assistant (aligned with generic safety objectives),
>   - A task-specialized expert that is better at maximizing the reward model but more brittle.
>   This creates a meaningful trade-off for the KL-based risk (stay near safe reference vs. exploit reward model).
>
> **Practical guidelines when you control expert training.**
>
> 1. **Reward diversity rather than randomness.**
>    Choose source tasks with **distinct reward vectors** so that their optimal policies occupy different regions of state–action space.
>
> 2. **Design for risk-relevant diversity.**
>    Anticipate likely deployment risks (hazards, overloads, behavioral drift) and ensure:
>    - some experts are risk-neutral/high-reward,
>    - some explicitly avoid dangerous regions,
>    - some mimic known-safe baselines.
>
> 3. **Avoid redundancy.**
>    Do not spend budget on many near-identical experts; they barely improve $\min_j \Delta_{\pi^\*}^{(T,j)}$.
>
> **Revision plan.**
> We will:
>
> - Make this connection to the localized mismatch term explicit in Section 4, and
> - Briefly describe in each experimental section how our expert construction follows these principles.

---

### Author Response · Authors · 2025-12-03

Dear AC and SAC,

We appreciate your time and the reviewers’ efforts, especially given the unusual reviewing circumstances this year. Below we briefly summarize (i) what the paper contributes, (ii) the main themes in the reviews, and (iii) how our rebuttal and planned revisions address them.

---

### 1. Contributions in brief

Our paper introduces **TRAM**, a framework for **test-time risk adaptation** when you have a *fixed pool of pre-trained, risk-neutral experts* and are given a **risk functional only at deployment**. The main ingredients are:

1. **Occupancy-based risk formulation.** We model risk as a functional $\rho(d_\pi)$ of discounted occupancies, which subsumes barrier-type, variance-based, and drift-type risks (e.g., KL to a safe expert). This lets us handle rich, trajectory-level safety criteria rather than just per-step costs.

2. **Inference-time mixture-of-experts policy.** At test time, TRAM selects actions via a **risk-adjusted Q**:
   $$
   \tilde{Q}_c^{\pi_j}(s,a) = Q_T^{\pi_j}(s,a) - c\,g_\rho^{\pi_j}(s,a)
   $$
   and picks the best expert/action pair. This requires no retraining of experts and can adapt to new risk specifications (and risk levels) on the fly.

3. **Theoretical guarantees.** We analyze the **suboptimality gap** between TRAM and the optimal risk-aware policy, showing it decomposes into:
   - a **coverage term** measuring how close the source rewards are to the target reward along optimal trajectories, and
   - a term linear in the risk weight $c$ (the “price of safety”).

   We also study a **minimax result** for training experts: for a fixed expert budget $N$, risk-neutral training maximizes coverage over unknown future risks compared to baking risk into training.

4. **Empirical evidence across three regimes.**
   - **MiniGrid (tabular):** full occupancy control and visualization to understand the mechanism and different risk functionals (barrier and variance-type).
   - **Reacher with SF-DQN (continuous state + SFs):** standard SF-based transfer benchmark where TRAM operates on neural successor features and barrier risks.
   - **LLM alignment (high-dimensional structured outputs):** TRAM on top of reward-model scores and KL risk to a safe model, with evaluation by GPT-4 preferences.

---

> ### Author Response · Authors · 2025-12-03
>
> ---
>
> ### 2. What the reviewers asked for and how we addressed it
>
> **(a) Motivation and setting (R2, R4).**
> Reviewers were initially concerned about a mismatch between our broad motivation (distribution shifts, changing dynamics) and the formal setting (shared dynamics, changing rewards/risks only).
>
> - In the rebuttal we **tightened the scope**: TRAM is explicitly about *deployment-time changes in reward/risk* under **fixed dynamics** and a **fixed expert pool**.
> - We gave concrete scenarios (industrial robotics with evolving safety rules; deployed LLM systems with changing alignment policies) where this regime is realistic and retraining on every new risk is not.
> - We explicitly acknowledged dynamics changes as future work and sketched how a bound with transition divergence would extend Theorem 4.1.
>
> R4’s follow-up comment explicitly notes that the **motivation is now clear** and that the earlier concern about “one-step risk” was resolved.
>
> ---
>
> **(b) Risk modeling, “one-step risk”, and occupancy (R3, R4).**
> Several comments questioned whether risk is only immediate and how occupancies are defined and computed.
>
> - We clarified that risk is always defined as a **functional of the discounted occupancy measure** $d_\pi$, and therefore inherently **multi-step**. Even when we write local contributions $g_\rho^{\pi}(s,a)$, these are defined so that their expectation under $d_\pi$ equals a *trajectory-level* risk (e.g., time in danger regions, variance of a cumulative signal, or KL drift).
> - We gave explicit formulas for $d_\pi$ and explained how we estimate it:
>   - **MiniGrid:** tabular or Monte Carlo discounted counts (no function approximator).
>   - **Reacher:** rollouts with simple geometric indicators (proximity to obstacles / limits), not a high-dimensional density model.
>   - **LLMs:** no occupancy is used; risk is KL drift at the sequence level.
> - We discussed OOD concerns for learned occupancy models and explained that in this paper we **avoid** that failure mode by only using on-policy estimates for fixed experts.
>
> R4’s follow-up acknowledges that the “one-step” concern was based on a misunderstanding and is now resolved.

---

> ### Author Response · Authors · 2025-12-03
>
> ---
>
> **(c) Successor features and SF-specific innovation (R3, R4).**
> A key concern was that much of the novelty appears in the SF/occupancy formulation, yet the LLM experiment does not use SFs.
>
> - We clarified that the **SF machinery is fully exercised in MiniGrid and especially in Reacher**:
>   - We added precise definitions of the feature map $\phi$, task vectors $w$, and SF-DQN architecture.
>   - We explained how Reacher’s multiple goal tasks induce a linear reward family where SFs give exact rewards and fast transfer.
> - For the **LLM experiment**, we deliberately do *not* use SFs because more direct and efficient Q-style estimators are available:
>   - We use a reward model and token-level scores to approximate “Q-like” values, as in value-guided decoding / Transfer-$Q^\*$-style methods.
>   - This fits transformers naturally and is more efficient than trying to build SFs over high-dimensional text states.
>
> We also committed to **reposition the LLM experiment** as a *secondary high-dimensional case study* (scalability of the mechanism) and to emphasize MiniGrid+Reacher as the **primary SF/occupancy evaluations** in the revised paper. R4’s latest comment indicates that clarity and soundness are significantly strengthened by these changes.
>
> ---
>
> **(d) Theory: usefulness, minimax claim, and transitions (R2, R3).**
>
> - We clarified that Theorem 4.1’s bound is primarily about **how suboptimality scales with reward mismatch and the risk weight**; even if the localized reward discrepancy is not directly computable, it is conceptually informative and matches the structure seen in related SF and transfer results.
> - We toned down the **“minimax” language**, making explicit that it applies to:
>   - a **fixed expert budget** $N$,
>   - a family of Lipschitz risks and shared dynamics, and
>   - procedures that train experts once and then use TRAM at test time.
>   Under these assumptions, risk-neutral expert training maximizes coverage in occupancy space; we do not claim global optimality beyond this model class.
> - We showed how a **transition-divergence term** could be added to the bound when dynamics differ, and clarified that our current analysis focuses on shared dynamics for clarity.
>
> ---
>
> **(e) Experiments: simplicity, baselines, and missing details (R1–R4).**
>
> 1. **Simplicity of MiniGrid and discretized Reacher; scalability.**
>    - We reframed MiniGrid and Reacher explicitly as **mechanism-demonstrating** environments:
>      - MiniGrid for fully inspectable tabular occupancies and risk visualizations.
>      - Reacher (a standard SF benchmark) for continuous state + SF-based value transfer with discrete actions for exact argmax.
>    - We emphasized that **scalability** is addressed by the **LLM alignment experiment**, where states are full dialogues, actions are sequences over a large vocabulary, and we evaluate with GPT-4 preferences.
>    - We acknowledged that richer continuous-control safety suites with non-discretized actions are valuable future work, but beyond what can be credibly added during rebuttal.
>
> 2. **Baselines, especially Safety-Constrained Policy Transfer with SFs.**
>    - We explained that SCPT-SF-style methods assume:
>      - known safety constraints at training time for the *target* task, and
>      - the ability to *train a new policy* on that task.
>    - TRAM targets the complementary regime: **fixed experts, unknown future risks, no retraining at deployment**.
>    - We committed to add a **dedicated related-work comparison** clarifying these differences in assumptions and deployment scenarios, and to treat a full SCPT-SF empirical comparison as a natural follow-up rather than a rushed addition.
>
> 3. **LLM experiment details and failure modes.**
>    - We gave a step-by-step description of:
>      - state/action spaces,
>      - candidate generation,
>      - reward-model scoring,
>      - KL risk computation,
>      - TRAM vs. baselines, and
>      - GPT-4 A/B evaluation and “GPT-4 win vs TRAM” metric.
>    - We discussed **failure modes**, e.g., when KL risk aligns TRAM with a suboptimal safe expert, and how this relates to reward hacking in the risk-free baseline.
>
> 4. **Hyperparameters, seeds, and statistical significance.**
>    - We committed to report and release code.
>
> 5. **Risk weight and number/diversity of experts.**
>    - We explained how the bound decomposes into:
>      - a linear “price of safety” in $c$, and
>      - a coverage term determined by expert diversity.
>    - We described our tuning protocol for $c$ and how expert diversity is enforced in our source tasks (different goals / reward weights in control; safe vs. high-reward experts in LLMs), and we committed to add small summary plots/tables to make these dependencies explicit.

---

> ### Author Response · Authors · 2025-12-03
>
> ### 3. Overall
>
> After the rebuttal and follow-up, several core concerns—**motivation, one-step risk, clarity of theory, and methodology**—have been explicitly acknowledged as resolved or significantly improved (e.g., by R4). The remaining points are largely about **positioning and experimental scope**:
>
> - better foregrounding MiniGrid+Reacher as the main SF/occupancy evaluations,
> - clearly positioning the LLM experiment as a secondary scalability case study, and
> - clarifying how TRAM relates to existing SF-based safe transfer methods like SCPT-SF.
>
> We have outlined concrete, targeted revisions addressing these issues without changing the core algorithm or results. We hope this meta-response helps you see how the reviews have shaped a clearer and more focused version of the paper, and we respectfully ask you to take these clarifications and commitments into account when making your decision.

---

### Meta-Review · Area_Chair_ZxhD · 2026-01-07

**Summary:**

The paper argues for a test-time adaptation framework in deploying reinforcement learning agents under safety constraints that are unknown during training. The proposed method pre-trains a set of risk-neutral source policies using successor features and estimates their occupancy measures. During test time, it combines the learned source policies into a mixture of experts by selecting actions that maximize a scalarized objective of task reward minus a penalty derived from a user-specified risk functional.

Multiple reviewers (u9yv, KKBM) questioned the exclusion of strong baselines like "relabeling + offline RL" or Safety-Constrained Policy Transfer (SCPT-SF). Reviewer u9yv mentioned a big difference between the proposed method and the experimental evaluation. The main technical contribution focuses on successor features but the LLM experiment ignores it for standard reward model scoring.  Reviewer KKBM also questioned that the introduction motivates the work via dynamics shifts and unexpected physical hazards but the formal problem statement assumes fixed dynamics where only the reward/risk functions change.

**Reviewer Concerns:**

Reviewer u9yv clearly mentioned that the paper require substantial additions and revisions. I do not think other reviewers would have changed their scores.

**Reviewer Scores:**

I do not think other reviewers would have changed their scores.

---

### Decision · Program_Chairs · 2026-01-26

Reject